# A NEW LOOK AT FAIRNESS IN STOCHASTIC MULTI-ARMED BANDIT PROBLEMS

## ABSTRACT

We study an important variant of the stochastic multi-armed bandit (MAB) problem, which takes fairness into consideration. Instead of directly maximizing cumulative expected reward, we need to balance between the total reward and fairness level. In this paper, we present a new insight in MAB with fairness and formulate the problem in the penalization framework, where rigorous penalized regret can be well defined and more sophisticated regret analysis is possible. Under such a framework, we propose a hard-threshold UCB-like algorithm, which enjoys many merits including asymptotic fairness, nearly optimal regret, better tradeoff between reward and fairness. Both gap-dependent and gap-independent upper bounds have been established. Lower bounds are also given to illustrate the tightness of our theoretical analysis. Numerous experimental results corroborate the theory and show the superiority of our method over other existing methods.

## 1 INTRODUCTION

The multi-armed bandit (MAB) problem is a classical framework for sequential decision-making in uncertain environments. Starting with the seminal work of Robbins (1952), over the years, a significant body of work has been developed to address both theoretical aspects and practical applications of this problem. In a traditional stochastic multi-armed bandit (MAB) problem (Lai & Robbins, 1985; Auer et al., 2002; Vermorel & Mohri, 2005; Bubeck & Cesa-Bianchi, 2012), a learner has access to $K$ arms and pulling arm $k$ generates a stochastic reward for the principal from an unknown distribution $F_k$ with an unknown expected reward $\mu_k$. If the mean rewards were known as prior information, the learner could just repeatedly pull the best arm given by $k^* = \arg\max_k \mu_k$. However, the learner has no such knowledge of the reward of each arm. Hence, one should use some learning algorithm $\pi$ which operates in rounds, pulls arm $\pi_t \in \{1, \dots, K\}$ in round $t$, observes the stochastic reward generated from reward distribution $F_{\pi_t}$, and uses that information to learn the best arm over time. The performance of learning algorithm $\pi$ is evaluated based on its cumulative regret over time horizon $T$, defined as

$$\bar{R}_\pi(T) = \mu_{k^*} T - \sum_{t=1}^{T} \mathbb{E}\mu_{\pi_t}. \tag{1}$$

To achieve the minimum regret, a good learner should make a balance between exploration (pulling different arms to get more information of reward distribution of each arm) and exploitation (pulling the arm currently believed to have the highest reward).

In addition to the above classical MAB problem, many variations of the MAB framework have been extensively studied in the literature recently. Various papers study MAB problems with additional constraints which include bandits with knapsack constraints (Badanidiyuru et al., 2013), bandits with budget constraints (Xia et al., 2015), sleeping bandits (Kleinberg et al., 2010; Chatterjee et al., 2017), etc. Except these, there is a huge research interest in fairness within machine learning field. Fairness has been a hot topic of many recent application tasks, including classification (Zafar et al., 2017a;b; Agarwal et al., 2018; Roh et al., 2021), regression (Berk et al., 2017; Rezaei et al., 2019), recommendation (Celis et al., 2017; Singh & Joachims, 2018; Beutel et al., 2019; Wang et al., 2021), resource allocation (Baruah et al., 1996; Talebi & Proutiere, 2018; Li et al., 2020), Markov decision process (Khan & Goodridge, 2019), etc. There are two popular definitions of fairness in the MAB literature. 1). The fairness is introduced into the bandit learning framework by saying that it is unfair

to preferentially choose one arm over another if the chosen arm has lower expected reward than the unchosen arm (Joseph et al., 2016). In other words, the learning algorithm cannot favor low-reward arms. 2). The fairness is introduced such that the algorithm needs to ensure that uniformly (i.e., at the end of every round) each arm is pulled at least a pre-specified fraction of times (Patil et al., 2020). In other words, it imposes an additional constraint to prevent the algorithm from playing low-reward arms too few times.

In this paper, we adopt a new perspective, e.g., in addition to maximizing the cumulative expected reward, it also allows the user to specify how "hard" or how "soft" the fairness requirement on each arm is. In this view, it is not always easy even to formulate the problem and to introduce an appropriate notion of regret. We thus propose a new formulation of fairness MAB by introducing penalty term $A_k \max(\tau_k T - N_k(T), 0)$, where $A_k, \tau_k$ are the penalty rate and fairness fraction for arm $k$ and $N_k(T)$ is the number of times pulling arm $k$. Hence it gives penalization when the algorithm fails to meet the fairness constraint and penalty term is proportional to the gap between pulling number and its required level. To solve this regularized MAB problem, we also propose a hard-threshold upper confidence bound (UCB) algorithm. It is similar to the classical UCB algorithm but adds an additional term to encourage the learner to favor those arms whose pulling numbers are below the required level at each round. The advantage of our approach is that it allows the user to distinguish , if desired, between arms for which is more important to sample an arm with required frequency and those arms for which it is less important to do so.

To the best of our knowledge, there is no work on mathematical framework of fairness MAB with regularization term in the literature. In this paper, we provide a relatively complete theory for the fairness MAB. We rigorously formalize the penalized regret which can be used for evaluating the performance of learning algorithm under fairness constraints. On theoretical side, the hard-threshold UCB algorithm is proved to achieve asymptotic fairness when a large penalty rate is chosen. The algorithm is shown to obtain $O(\log T)$ regret when the sub-optimality gap is assumed to be fixed. Additionally, the characterization of fluctuation of non-fairness level, $\max_{1 \leq t \leq T} \max(\tau_k t - N_k(t), 0)$ is also given. Its magnitude is also shown to be $O(\log T)$. Moreover, we establish a sub-optimal gap-free regret bound of proposed method and provide insights on how hard-threshold based UCB index works. We also point out that the analysis of proposed hard-threshold UCB algorithm is much harder than the classical UCB due to the existence of interventions between different sub-optimal arms. On numerical side, the experimental results confirm our theory and show that the performance of the proposed algorithm is better than other popular methods. Our method achieves a better trade-off between reward and fairness.

**Notations**. For real number $x$, $(x)_+$ stands for $\max\{0, x\}$; $\lfloor x \rfloor$ is the largest integer smaller or equal to $x$. For integer $n$, we use $[n]$ to represent the set $\{1, \ldots, n\}$. We say $a = O(b)$; $a = \Omega(b)$ if there exists a constant $C$ such that $a \leq Cb$; $a \geq b/C$. The symbols $\mathbb{E}$ and $\mathbb{P}(\cdot)$ denote generic expectation and probability under a probability measure that may be determined from the context. We let $\pi$ be a generic policy / learning algorithm.

## 2 ACHIEVING FAIRNESS VIA PENALIZATION

Consider a stochastic multi-armed bandit problem with $K$ arms and unknown expected rewards $\mu_1, \ldots, \mu_K$ associated with these arms. The notion of fairness we introduce consists of proportions $\tau_k \geq 0$, $k = 1, \ldots, K$ with $\tau_1 + \cdots + \tau_K < 1$. We use $T \in \{1, 2, \ldots, \}$ to denote the time horizon and $N_{k,\pi}(t)$ to denote the number of times that arm $k$ has been pulled by time $t \in [T]$ using policy $\pi$. For notational simplicity, we may write $N_{k,\pi}(t)$ as $N_k(t)$. It is desired to pull arm $k$ at least at the uniform rate of $\tau_k$, $k = 1, \ldots, K$. In other words, the learner should obey the constraint that $N_k(t) \geq \tau_k t$ for any $t \in [T]$. Thus a good policy aims to solve the following optimization problem,

$$\arg\max_{\pi} \mathbb{E} \sum_k \mu_k N_{k,\pi}(T), \text{ subject to } N_{k,\pi}(t) \geq \tau_k t \text{ for all } k \text{ and } t. \tag{2}$$

Instead of directly working with such a constrained bandit problem, we consider a penalization problem. That is, one gets penalized if the arm is not pulled sufficiently often. To reflect this, we introduce the following design problem. Let $S_\pi(T)$ be the sum of the rewards obtained by time $t$ under policy $\pi$, i.e., $S_\pi(T) = \sum_{t=1}^{T} r_{\pi_t}$ where $\pi_t$ is the arm index chosen by policy $\pi$ at time

$t \in [T]$ and $r_{\pi_t}$ is the corresponding reward. Then the penalized total reward is defined as

$$S_{\text{pen},\pi}(T) = S_\pi(T) - \sum_{k=1}^{K} A_k \big(\tau_k T - N_{k,\pi}(T)\big)_+, \tag{3}$$

where $A_1, \ldots, A_K$ are known nonnegative penalty rates. Our goal is to design a learning algorithm to make the expectation of $S_{\text{pen},\pi}(T)$ as large as possible. By taking the expectation, we have

$$\mathbb{E}[S_{\text{pen},\pi}(T)] = \sum_{k=1}^{K} \mu_k \mathbb{E}[N_{k,\pi}(t)] - \sum_{k=1}^{K} A_k \mathbb{E}[(\tau_k T - N_{k,\pi}(T))_+], \tag{4}$$

which is the penalized reward achieved by policy $\pi$ and we would like to maximize it over $\pi$. Now we are ready to introduce the *penalized regret* function, which is the core for the regret analysis.

To derive the new regret, we first note that maximizing $\mathbb{E}[S_{\text{pen},\pi}(T)]$ is the same as minimizing the following loss function,

$$L(T) = \mu^* T - \mathbb{E}[S_{\text{pen},\pi}(T)] = \sum_{k=1}^{K} \Big[ \Delta_k \mathbb{E}[N_k(t)] + A_k \mathbb{E}[(\tau_k T - N_k(T))_+] \Big], \tag{5}$$

where we denote

$$\mu^* = \max_{k=1,\ldots,K} \mu_k, \ \Delta_k = \mu^* - \mu_k, \ k = 1, \ldots, K.$$

In order to find the minimum possible value of $L(T)$, let us understand what a prophet (who knows the expected rewards $\mu_1, \ldots, \mu_K$) would do. Clearly, a prophet (who, in addition, is not constrained by integer value) would solve the following optimization problem,

$$\min_{x_1,\ldots,x_K} \sum_{k=1}^{K} \Big[ \Delta_k x_k + A_k \mathbb{E}(\tau_k T - x_k)_+ \Big] \text{ subject to } \sum_{k=1}^{K} x_k = T, \ x_k \geq 0, \ k = 1, \ldots, K,$$

and pull arm $k$ for $x_k$ times ($k = 1, \ldots, K$). By denoting $y_k = x_k/T, \ k \in [K]$, we transform this problem into

$$\min_{y_1,\ldots,y_K} \sum_{k=1}^{K} \Big[ \Delta_k y_k + A_k (\tau_k - y_k)_+ \Big] \text{ subject to } \sum_{k=1}^{K} y_k = 1, \ y_k \geq 0, \ k = 1, \ldots, K. \tag{6}$$

We will solve the problem (6) by finding $y_1, \ldots, y_K$ that satisfy the constraints and that minimize simultaneously each term in the objective function. It is not hard to observe the following facts.

1. For $A \geq 0$, function $y \mapsto A(\tau - y)_+$ achieves its minimum value of 0 for $y \geq \tau$.
2. For $A \geq \Delta > 0$, function $y \mapsto \Delta y + A(\tau - y)_+$ achieves its minimum of $\Delta\tau$ at $y = \tau$.
3. For $\Delta > A \geq 0$, function $y \mapsto \Delta y + A(\tau - y)_+$ achieves its minimum of $A\tau$ at $y = 0$.

As a result, we introduce the following three sets

$$\mathcal{A}_{\text{opt}} = \big\{ k \in [K] : \mu_k = \mu^* \big\}, \mathcal{A}_{\text{cr}} = \big\{ k \in [K] : A_k \geq \Delta_k > 0 \big\}, \mathcal{A}_{\text{non-cr}} = \big\{ k \in [K] : \Delta_k > A_k \big\},$$

where $\mathcal{A}_{\text{opt}}$ consists of all optimal arms, $\mathcal{A}_{\text{cr}}$ consists of sub-optimal arms with penalty rate larger than (or equal to) the sub-optimal gap and $\mathcal{A}_{\text{non-cr}}$ includes sub-optimal arms with penalty rate smaller than the sub-optimal gap. Therefore an optimal solution to the problem (6) can be constructed as follows. Let $k^*$ be an arbitrary arm in $\mathcal{A}_{\text{opt}}$, and choose

$$y_k = \begin{cases} 1 - \sum_{j \in \mathcal{A}_{\text{cr}} \cup (\mathcal{A}_{\text{opt}} \setminus \{k^*\})} \tau_j, & k = k^*, \\ \tau_k, & k \in \mathcal{A}_{\text{cr}} \cup (\mathcal{A}_{\text{opt}} \setminus \{k^*\}), \\ 0, & k \in \mathcal{A}_{\text{non-cr}}. \end{cases} \tag{7}$$

Therefore, a prophet would choose (modulo rounding) in (5)

$$N_k(T) = \begin{cases} \Big(1 - \sum_{j \in \mathcal{A}_{\text{cr}} \cup (\mathcal{A}_{\text{opt}} \setminus \{k^*\})} \tau_j \Big) T, & k = k^*, \\ \tau_k T, & k \in \mathcal{A}_{\text{cr}} \cup (\mathcal{A}_{\text{opt}} \setminus \{k^*\}), \\ 0, & k \in \mathcal{A}_{\text{non-cr}}, \end{cases} \tag{8}$$

leading to the following optimal value of $L(T)$,

$$L^*(T) = \sum_{k \in \mathcal{A}_{\mathrm{cr}}} \Delta_k \tau_k T + \sum_{k \in \mathcal{A}_{\mathrm{non-cr}}} A_k \tau_k T = \left( \sum_{k=1}^K \min(\Delta_k, A_k) \tau_k \right) T. \tag{9}$$

Given an arbitrary algorithm $\pi$, we can therefore define the penalized regret as

$$R_\pi(T) = L_\pi(T) - L^*(T) = \sum_{k \in \mathcal{A}_{\mathrm{opt}}} A_k \mathbb{E}\big(\tau_k T - N_{k,\pi}(T)\big)_+ \tag{10}$$

$$+ \sum_{k \in \mathcal{A}_{\mathrm{cr}}} \Big[ \Delta_k \mathbb{E}\big(N_{k,\pi}(T) - \tau_k T\big) + A_k \mathbb{E}\big(\tau_k T - N_{k,\pi}(T)\big)_+ \Big]$$

$$+ \sum_{k \in \mathcal{A}_{\mathrm{non-cr}}} \Big[ \Delta_k \mathbb{E} N_{k,\pi}(T) + A_k \Big( \mathbb{E}\big(\tau_k T - N_{k,\pi}(T)\big)_+ - \tau_k T \Big) \Big].$$

## 3 A HARD-THRESHOLD UCB ALGORITHM

We now introduce a UCB-like algorithm which aims to achieve the minimum penalized regret described in the previous section. We assume that all rewards take values in the interval $[0, 1]$. We denote by $X_n^{(k)}$ the reward obtained after pulling arm $k$ for the $n$th time, $k \in [K]$, $n = 1, 2, \ldots$. Let

$$\hat{m}_k(n) = \frac{1}{N_n(k)} \sum_{i=1}^{N_n(k)} X_i^{(k)}, \ k = 1, \ldots, K, \ n = 1, 2, \ldots \tag{11}$$

and introduce the following index: for $k = 1, \ldots, K, \ n = 1, 2, \ldots$ set

$$i_k(n) = \hat{m}_k(n-1) + A_k \mathbf{1}\big(N_k(n-1) < \tau_k n\big) + \sqrt{\frac{2 \log n}{N_k(n-1)}}. \tag{12}$$

The algorithm proceeds as follows. It starts by pulling each arm once. Then at each subsequent step, we pull an arm with the highest value of the index $i_k(n)$. In equation 12, there is an additional term $A_k \mathbf{1}(N_k(n-1) < \tau_k n)$ compared with classical UCB algorithm. It takes the hard threshold form. Once the number of times that arm $k$ has been pulled before time $n$ is less than the *fairness level* ($\tau_k n$) at round $n$, penalty rate $A_k$ will be added to the UCB index. In other words, the proposed algorithm favors those arms which does not meet the fairness requirement. The detailed implementation is given in Algorithm 1.

## 4 THEORETICAL ANALYSIS OF THE HARD-THRESHOLD UCB ALGORITHM

In this section, we present theoretical results for the hard-threshold UCB algorithm introduced in Section 3. Throughout this section, we need to introduce additional notation and concepts. We say $\tau_k = \tilde{\Omega}(1)$ if it is a positive constant which is independent of $T$. We assume that there exists a positive constant $c_0$ such that $\sum_k \tau_k \leq 1 - c_0$ and $T$ is much larger than $K$. The penalty rates $A_k$'s are assumed to be known fixed constants. The expected reward $\mu_k$ ($k \in [K]$) is assumed between 0 and 1. Hence sub-optimality gap $\Delta_k$ is between 0 and 1 as well.

**Asymptotic Fairness**. Given the large penalty rates, the proposed algorithm can guarantee the asymptotic fairness for any arm $k \in [K]$. In other words, the algorithm can guarantee that the number of times that arm $k$ has been pulled up to time $T$ is at least $\lfloor \tau_k T \rfloor$ with high probability.

**Theorem 1** *If $A_k - \Delta_k \geq \min\{\sqrt{\frac{32 \log T}{\tau_k T}}, 1\}$ and $\tau_k = \tilde{\Omega}(1)$ for all $k$, we have $N_k(T) \geq \lfloor \tau_k T \rfloor$ for any $k$ with probability going to 1 as $T \to \infty$.*

Theorem 1 tells us that the proposed algorithm treats every arm fairly when the penalty rate dominates the sub-optimality gap.

---

**Algorithm 1** Hard-Threshold UCB Algorithm.

---

1: **Input.** Number of arms $K$, fairness proportions $\tau_k$'s, penalty rates $A_k$'s, time horizon $T$.
2: **Output.** Cumulative reward, the number of times that each arm is played ($N_k(T)$, $k \in \{1, \ldots, K\}$.)
3: **Initialization.**
   For each $k \in \{1, \ldots, K\}$, we set initial count $N_k = 0$ and arm-specific cumulative reward $R_k = 0$.
4: **while** $n \leq T$ **do**
5:    If $n \leq K$, we choose $k_n = n$.
6:    If $n > K$, we choose $k_n = \arg\max_k i_k(n)$.
7:    We observe reward $r_n$. We update count $N_{k_n} = N_{k_n} + 1$ and update reward $R_{k_n} = R_{k_n} + r_n$.

8:    We update hard-threshold index for each arm $k \in \{1, \ldots, K\}$ by calculating

$$i_k(n+1) = R_k/N_k + A_k \mathbf{1}(N_k < \tau_k(n+1)) + \sqrt{\frac{2\log n}{N_k}}.$$

9:    Increase time index $n$ by 1.
10: **end while**
11: Return vector $(N_k)$.

---

### 4.1 REGRET ANALYSIS: UPPER BOUNDS

In this section, we provide upper bounds on the penalized regret defined in equation 10 under two scenarios. (1) We establish the gap-dependent bound when the sub-optimality $\Delta_k$'s are fixed constants. (2) We prove the gap-independent bound when $\Delta_k$'s vary within the interval $[0, 1]$.

**Theorem 2** *(Gap-dependent Upper bound.) Assume that $A_k - \Delta_k \geq c_a$ ($c_a$ is a positive constant) holds for any arm $k \in \mathcal{A}_{opt} \cup \mathcal{A}_{cr}$ and $\Delta_k - A_k \geq \sqrt{\frac{8K\log T}{c_0^2 T}}$ holds for any $k \in \mathcal{A}_{non\text{-}cr}$. We then have the following results.*

*For any $k \in \mathcal{A}_{opt} \cup \mathcal{A}_{cr}$, it holds*

$$\mathbb{E}[(\tau_k T - N_k(T))_+] = O(1).$$

*For any $k \in \mathcal{A}_{cr}$, it holds*

$$\mathbb{E}[N_k(T)] \leq \max\{\frac{8\log T}{\Delta_k^2}, \tau_k T\} + O(1).$$

*For any arm $k_j \in \mathcal{A}_{non\text{-}cr}$, it holds*

$$\mathbb{E}[N_k(T)] \leq \max\{\min\{\frac{8\log T}{(\Delta_k - A_k)^2}, \tau_k T\}, \frac{8\log T}{\Delta_k^2}\} + O(1).$$

*Therefore, we have*

$$R_\pi(T) \leq \sum_{k \in \mathcal{A}_{cr}} (\frac{8\log T}{\Delta_k} - \tau_k T)_+ + \sum_{k \in \mathcal{A}_{non\text{-}cr}} \max\{\min\{\frac{8\log T}{\Delta_k - A_k}, (\Delta_k - A_k)\tau_k T\}, \frac{8\log T}{\Delta_k}\} + O(K). \quad (13)$$

Theorem 2 tells us that the number of times that each arm $k$ in critical set $\mathcal{A}_{cr}$ is played is at least around fairness requirement $\tau_k T$ when the the penalty rate is larger than the sub-optimality gap by some constant. On the other hand, for each arm $k$ in non-critical set $\mathcal{A}_{non\text{-}cr}$, it could be played less than fairness requirement when sub-optimality gap substantially dominates the penalty rate. The total penalized regret has order of $\log T$ and is hence nearly not improvable. In addition, when $A_k \equiv 0$ and it degenerates to the classical settings, then all arms become non-critical arms and our bound reduces to $O(\sum_k \frac{8\log T}{\Delta_k})$ which matches the existing result (Auer et al., 2002).

**Maximal Inequality**. In Theorem 2 above we have shown that $\mathbb{E}[(\tau_k T - N_k(T))_+] = O(1)$ for any $k \in \mathcal{A}_{opt} \cup \mathcal{A}_{cr}$ under mild conditions on $\Delta_k$'s. In the result below, we derive a maximal inequality for the *non-fairness level*, $(\tau_k t - N_k(t))_+$, $t \in [T]$.

**Theorem 3** *Order the $K$ arms in such a way that*

$$A_{k_1} + \mu_{k_1} \geq \ldots \geq A_{k_j} + \mu_{k_j} \geq \ldots \geq A_{k_K} + \mu_{k_K}.$$

*Then for any arm $k_j \in \mathcal{A}_{opt} \cup \mathcal{A}_{cr}$, we have*

$$\mathbb{E}[\max_{1 \leq t \leq T} (\tau_{k_j} t - N_{k_j}(t))_+] \leq a_j \log T + O(1),$$

*where the coefficient $a_j$ is defined as*

$$a_j = 8 \sum_{d=1}^{j} (j - d + 1) \left( \sum_{m=1}^{d-1} \frac{1}{(\mu_{k_d} + A_{k_d} - \mu_{k_m})^2} + \sum_{m=d+1}^{K} \frac{1}{(\mu_{k_d} + A_{k_d} - \mu_{k_m} - A_{k_m})^2} \right). \tag{14}$$

Theorem 3 nearly guarantees the ANY-ROUND fairness for all arms $k \in [K]$ up to a $O(\log T)$ difference.

**Gap-independent Upper bound**. We now switch to establishing a gap-independent upper bound. It relies on the following observations. The key challenge is how to bound the term $\mathbb{E}[(\tau_k T - N_k(T))_+]$ for $k \in \mathcal{A}_{opt} \cup \mathcal{A}_{cr}$.

(*Observation 1*) If $A_k - \Delta_k \leq 4\sqrt{\frac{2 \log T}{T^{2/3}}}$, then $(A_k - \Delta_k)\mathbb{E}[(\tau_k T - N_k(T))_+] \leq 4\tau_k T^{2/3}(2 \log T)^{1/2}$.

**Lemma 1** *(Observation 2)  If arm $k$ satisfies that $A_k - \Delta_k \geq 4\sqrt{\frac{2 \log T}{T^{2/3}}}$ and $\tau_k = \tilde{\Omega}(1)$, then we have $\mathbb{E}[(\tau_k T - N_k(T))_+] = O(\tau_k K T^{2/3})$.*

Based on above observations, we have the following regret bound.

**Theorem 4** *When $\tau_k = \tilde{\Omega}(1)$, it holds that*

$$R_\pi(T) \leq 8\sqrt{T \log T}(\sum_k \sqrt{\tau_k}) + 8\sqrt{(1 - \tau_{min})KT \log T} + A_{max}KT^{2/3}(2 \log T)^{1/2}, \tag{15}$$

*where $\tau_{min} = \min_k \tau_k$ and $A_{max} = \max_k A_k$.*

The first term in (15) is for $A_k(\mathbb{E}(\tau_k T - N_k(T))_+)$ with $k \in \mathcal{A}_{non-cr}$. The second term gives a bound for $\Delta_k \mathbb{E}(N_k(T) - \tau_k T)$ for $k \in [K]$. The third term in (15) is for bounding $A_k \mathbb{E}(\tau_k T - N_k(T))_+$ for $k \in \mathcal{A}_{opt} \cup \mathcal{A}_{cr}$.

## 4.2 REGRET ANALYSIS: LOWER BOUNDS

**Gap-dependent Lower Bound.**   In this part, we first show that the bound given in inequality (13) is tight. To see this, the results are stated in the following theorems.

**Theorem 5** *There exists a bandit setting for which the regret of proposed algorithm has the following lower bound,*

$$R_\pi(T) \geq \sum_{k \in \mathcal{A}_{non-cr}, \tau_k > 0} \frac{\log T}{\Delta_k - A_k}. \tag{16}$$

**Theorem 6** *There exists a bandit setting for which the regret of proposed algorithm has the following lower bound,*

$$R_\pi(T) \geq \sum_{k \in \mathcal{A}_{cr}} \Delta_k (\frac{\log T}{\Delta_k^2} - \tau_k T). \tag{17}$$

Theorem 5 says that the term $\log T/(\Delta_k - A_k)$ is nearly optimal up a multiplicative constant 8 for any arm in the non-critical set. Similarly, Theorem 6 tells us that $(\frac{8 \log T}{\Delta_k} - \tau_k T)_+$ is also nearly

optimal for arms in the critical set. Therefore, Theorem 2 gives a relatively sharp gap-dependent upper bound. It is almost impossible to improve the regret bound analysis for our proposed hard-threshold UCB algorithm in the instance-dependent scenario.

**Gap-independent Lower Bound**. We also obtain a gap-independent lower bound as follows.

**Theorem 7** *Let $K > 1$ and $T$ be a large integer. Penalty rates $A_1, A_2, \ldots, A_K$ are fixed positive constants. Assume that the fairness parameters $\tau_1, \ldots, \tau_K \in [0, 1]$ with $\sum_k \tau_k < 1$. Then, for any policy $\pi$, there exists a mean vector $\boldsymbol{\mu} = (\mu_1, \ldots, \mu_K)$ such that*

$$R_\pi(T) \geq C(1 - 2\max_k \tau_k)\sqrt{(K-1)T},$$

*where $C$ is a universal constant which does not depend on $A_k$, $\tau_k$'s.*

By comparing Theorems 4 and 7, we can see that there is a substantial gap. This is because term $A_k \mathbb{E}[(\tau_k T - N_k(T))_+]$ is very hard to handle. This term can be trivially lower bounded below by zero for any algorithm. However, this term is proved to be $O(T^{2/3})$ (ignoring $\log T$ factor) by our current techniques under the proposed algorithm. Whether we can improve the gap-independent upper bound to be $O(T^{1/2})$ is an open question in the future work.

## 5   COMMENTS ON THE HARD-THRESHOLD UCB ALGORITHM

**On hard threshold**. In the proposed algorithm, we use a hard-threshold term $A_k \mathbf{1}(N_k(n-1) < \tau_k n)$ in constructing a UCB-like index $i_k(n)$. A natural question is whether we can use a soft-threshold index by defining

$$\tilde{i}_k(n) = \hat{m}_k(n-1) + A_k \frac{\max(\tau_k n - N_k(n-1), 0)}{\tau_k n} + \sqrt{\frac{2\log n}{N_k(n-1)}}?$$

The answer is negative in the sense that $\tilde{i}_k(n)$ becomes a continuous function of $N_k$ and does not have a jump point at the critical value $\tau_k n$. Hence it does not give sufficient penalization to those arms $k$ which are below the fairness proportion $\tau_k$. Hence, a soft-threshold UCB-like index fails to guarantee the asymptotic fairness and nearly-optimal penalized regret.

**When $\tau_k$ is not constant**. In our theoretical analysis, we only consider the case that $\tau_k = \tilde{\Omega}(1)$ for ease of presentation. The current results could also apply when threshold $\tau_k$ is dependent on time horizon $T$ with $\tau_k(T) = 1/T^b (0 < b < 1)$.

**Technical Challenges**. Since the index $i_k(n)$ is a discontinuous function of $N_k$, this brings additional difficulties in analyzing the regret bound. The most distinguished feature from the classical regret analysis is that we cannot analyze term $N_k(T)$ separately for each sub-optimal gap $k$. In fact, the optimal arm ($\arg\max_k \mu_k$) is fixed for all rounds in the classical setting. In contrast, the "optimal arm" ($\arg\max_k \mu_k + A_k \mathbf{1}\{N_k < \tau_k n\}$) varies as the algorithm progresses in our framework. Due to such interventions among different arms, term $(\tau_k T - N_k(T))_+$ should be treated carefully.

**Connections to LASSO problems**. We would like to point out that our current framework shares similarities with LASSO problem (Tibshirani, 1996; Zhao & Yu, 2006; Zou, 2006) in linear regression models. Both of them introduces the penalization terms to enforce the solution to obey fairness constraints / sparsity to some degree. In our penalized MAB framework, whether an arm $k$ is played at least $\tau_k T$ times or not depends on the penalty rate $A_k$ and the sub-optimality gap $\Delta_k$. Similarly, in the LASSO framework, whether a coefficient is to be estimated as zero depends on the penalty parameter and its true coefficient value.

**Comparison with Baselines**. We compare the proposed methods with related existing methods.

*Learning with Fairness Guarantee (LFG, Li et al. (2019))*. It is implemented via following steps.

- For each round $n$, we compute the index for each arm, $\bar{i}_k(n) = \min\{\hat{m}_k(n-1) + \sqrt{\frac{2\log n}{N_k(n-1)}}, 1\}$ and compute queue length for each arm, $Q_k(n) = \max\{Q_k(n-1) + \tau_k - \mathbf{1}\{\text{arm } k \text{ is pulled}\}, 0\}$.

- The learner plays the arm which maximizes $Q_k(n) + \eta_0 w_k i_k(n)$ and receive the corresponding reward, where $\eta_0$ is the tuning parameter and $w_k$ is the known weight. Without loss of generality, we assume $w_k \equiv 1$ by treating each arm equally when we have no additional information.

*Fair-Learn (Flearn, Patil et al. (2020)).* Its main procedure is given as below.

- For each round $n$, we compute set $\mathcal{A}(n)$, $\mathcal{A}(n) := \{k : \tau_k(n-1) - N_k(n-1) > \alpha\}$, which contains those arms which are not fair at round $n$ at level.
- If $\mathcal{A}(n) \neq \emptyset$, we play arm which maximizes $\tau_k(n-1) - N_k(n-1)$. Otherwise, we play arm which maximizes $\hat{m}_k(n-1) + \sqrt{\frac{2\log n}{N_k(n-1)}}$.

Fair-learn method can enforce each arm $k$ should be played at proportion level $\tau_k$ only when $\alpha = 0$. LFG method fails to guarantee the asymptotic fairness when $\eta_0 > 0$. Neither of these methods can well balance between total rewards and fairness constraint as our method does.

## 6 EXPERIMENT RESULTS

A In this experimental setting, we examine the relationship between number of times that non-critical arm $k$ has been pulled at $T$ (=20000) rounds and the inverse gap $1/(\Delta_k - A_k)^2$. In particular, we construct the following three parameter settings ($\tau_k \equiv 1/20$).
Case 1: $K = 9$; $\boldsymbol{\mu} = (0.9, 0.8, 0.7, 0.6, 0.6, 0.4, 0.3, 0.2, 0.1)$; $A_k \equiv 0.45$.
Case 2: $K = 9$; $\boldsymbol{\mu} = (0.95, 0.8, 0.7, 0.6, 0.6, 0.4, 0.3, 0.2, 0.1)$; $A_k \equiv 0.41$.
Case 3: $K = 9$; $\boldsymbol{\mu} = (0.9, 0.8, 0.7, 0.6, 0.6, 0.425, 0.4, 0.375, 0.35)$; $A \equiv 0.45$.

B Similarly, we examine the relationship between number of times that critical arm $k$ has been pulled at $T$ (=20000) rounds and the inverse gap $1/\Delta_k^2$. We set $\tau_k \equiv 1/20$, $A_k \equiv 0.45$.
Case 1: $K = 9$; $\boldsymbol{\mu} = (0.9, 0.86, 0.84, 0.82, 0.6, 0.4, 0.3, 0.2, 0.1)$.
Case 2: $K = 9$; $\boldsymbol{\mu} = (0.95, 0.85, 0.84, 0.83, 0.82, 0.4, 0.3, 0.2, 0.1)$.
Case 3: $K = 9$; $\boldsymbol{\mu} = (0.9, 0.8, 0.7, 0.6, 0.5, 0.4, 0.3, 0.2, 0.1)$.

C We investigate the relationship between cumulative penalized regret and total time horizon ($T$) under three algorithms (proposed method, LFG, and Flearn). The parameters are constructed as follows. The number of arms ($K$) is set to be 5 or 20. The total time horizon ($T$) varies from 500 to 16000. The fairness proportion $\tau_k$ of each arm is set to be $\tau_k = \tau/K$ with $\tau \in \{0.2, 0.4, 0.8\}$. The penalty rate $A_k$ is constructed as $A_k \equiv (\max_k \mu_k - \min_k \mu_k)/2$. Each entry of the mean reward vector ($\mu_k$) is randomly generated between $[0, 1]$. The reward distribution of each arm is a Gaussian distribution, e.g., $N(\mu_k, \frac{1}{K^2})$. For Flearn algorithm, we take tuning parameter $\alpha = 0$. For LFG algorithm, we take $\eta_0 = \sqrt{T}$. Each case is replicated for 50 times.

From Figure 1, we can see that the pulling number $N_k(T)$ is proportional to $1/(\Delta_k - A_k)^2$ for $k \in \mathcal{A}_{\text{non-cr}}$ when $N_k(T)$ does not reach fairness level $\tau_k T$. We also see that $N_k(T)$ is proportional to $1/\Delta_k^2$ for $k \in \mathcal{A}_{\text{cr}}$ when the pulling number is larger than fairness level $\tau_k T$. These phenomena match the results in Theorem 2. From Figure 2, we observe that the proposed method achieves smaller penalized regret compared with LFG and Flearn. This confirms that our method is indeed a good learning algorithm under penalization framework.

In the appendix, we also study the paths of unfairness level $((\tau_k T - N_k(T))_+)$ when tuning parameter varies and investigate the relationship between total expected reward ($\sum_{t=1}^{T} \mu_{\pi_t}$) and unfairness level ($\sum_{k \in [K]} (\tau_k T - N_k(T))_+$) for three algorithms. From Figure 3 (See Appendix A), the paths of unfairness level show different behaviors under three algorithms. For our method, with scale parameter decreasing, each arm becomes unfair one by one. By contrast, all arms under both Flearn and LFG methods suddenly become unfair once scale parameter decreases from 1. This suggests that our method has sparsity feature as LASSO does, e.g., making arms with small sub-optimality gap fair. From Figure 4 (See Appendix A), we can tell that the proposed method always achieves the highest reward given the same unfairness level under different parameter settings. This gives evidence that hard-threshold UCB algorithm makes better balance between total reward and fairness constraints compared with other competing methods.

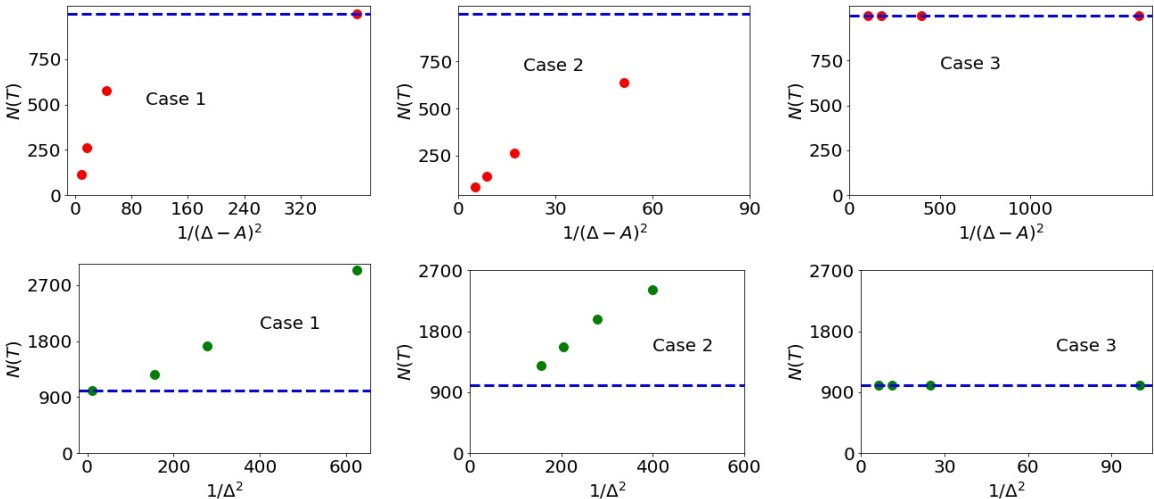

Figure 1: Upper row: $N_k(T)$ vs $1/(\Delta_k - A_k)^2$ for arm $k \in \mathcal{A}_{\text{non-cr}}$. Bottom row: $N_k(T)$ vs $1/\Delta_k^2$ for arm $k \in \mathcal{A}_{\text{cr}}$. In all plots, the blue horizontal line stands for fairness level $\tau_k T$.

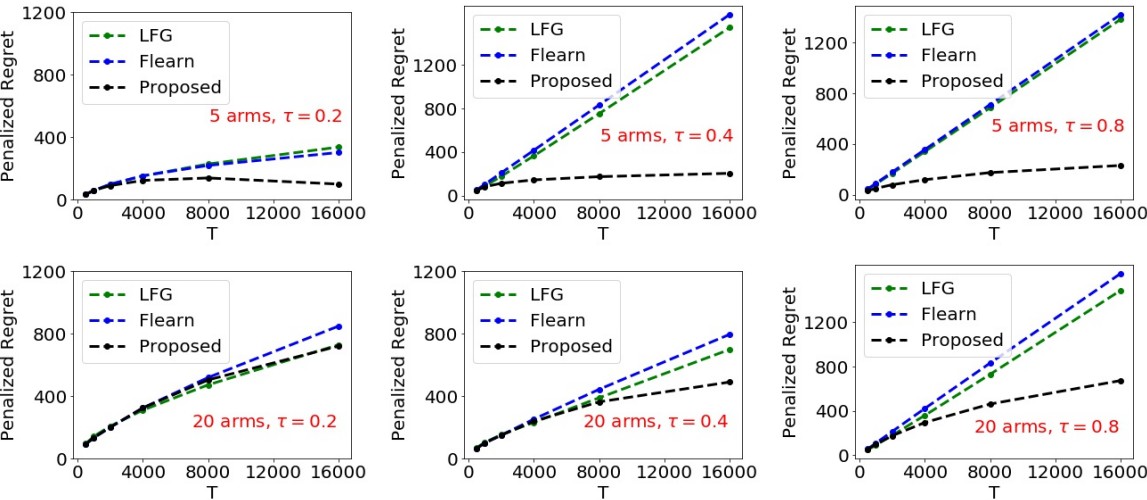

Figure 2: Penalized Regret ($R_\pi(T)$) vs Different Time Horizon ($T$) under different settings.

## 7 CONCLUSION

In this paper, we provide a new framework of fairness MAB problem by introducing regularization terms. The advantage of our new approach is that it allows the user to distinguish between arms for which is more important to sample an arm with required frequency level and arms for which it is less important to do so. A hard-threshold UCB algorithm is proposed and is shown to have good performance under this framework. Unlike other existing algorithms, the proposed algorithm not only achieves the asymptotic fairness but also handles well in balance between reward and fairness constraints. A relatively complete theory, including both gap-dependent / independent bounds, has been established. The new theoretical results contribute to the fairness in machine learning field and bring better insights in how to play smartly in the exploitation and exploration games.

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

## APPENDICES

In this appendix, the first section is dedicated for experimental results of Experiments D and E. In the rest, we collect all technical proofs. Specifically, the proofs of gap-dependent upper and lower bounds are given in Section B and C. The proofs of gap independent upper and lower bounds are given in Section E and F, respectively. The proof of $\mathbb{E}[\max_{1 \le t \le T}(\tau_k t - N_k(t))]_+$ is given in Section D.

## A THE PLOTS FOR EXPERIMENTS D AND E

D We investigate the path of unfairness level $((\tau_k T - N_k(T))_+)$ of each arm when the tuning parameter varies. The parameters of two settings are constructed as follows.

Setting 1: $K = 8, T = 10000; (\mu_1, \ldots, \mu_8) = (0.9, 0.7, 0.6, 0.5, 0.4, 0.3, 0.2, 0.1); \tau_1 = \ldots = \tau_8 = \frac{1}{2K}$. The reward distribution of each arm is a Gaussian distribution, e.g., $N(\mu_k, \frac{1}{K^2})$.

Setting 2: $K = 8, T = 10000; (\mu_1, \ldots, \mu_8) = (0.95, 0.7, 0.65, 0.6, 0.2, 0.15, 0.1, 0.05); \tau_1 = \ldots = \tau_4 = 0.8\frac{1}{K}$ and $\tau_5 = \ldots = \tau_8 = 0.4\tau_1$. Again, the reward distribution of each arm is a Gaussian distribution, e.g., $N(\mu_k, \frac{1}{K^2})$.

The penalty rates $A_k \equiv \eta$, where we call $\eta$ is the scale parameter which takes value between 0 and 1. For Flearn algorithm, the tuning parameter $\alpha = (1 - \eta)\tau_1 T$ with $\eta$ varying from 0 to 1. For LFG algorithm, the tuning parameter $\eta_0 = (1 - \eta)T$ with $\eta \in (0, 1]$.

When scale parameter $\eta \to 1$, three algorithms will prefer to exploit the arm with highest reward and pay less attention to the fairness. On the other hand, $\eta \to 0$, three algorithms tend to treat the fairness as the priority. (Due to space limit, the results are in Appendix A)

E We investigate the relationship between total expected reward $(\sum_{t=1}^{T} \mu_{\pi_t})$ and unfairness level $(\sum_{k \in [K]}(\tau_k T - N_k(T))_+)$ for three algorithms. The parameters are given as follows. We set $K \in \{5, 20\}$ and $\tau_k \equiv \tau/K$ with $\tau \in \{0, 20.5\}$. Each element in mean reward vector $(\mu_k)$ is generated between 0 and 1. Moreover, we generate the reward from three different distributions, (1) Gaussian $N(\mu_k, \frac{1}{K^2})$, (2) Beta $Beta(\mu_k, 1 - \mu_k)$, (3) Bernoulli $Bern(1, \mu_k)$.

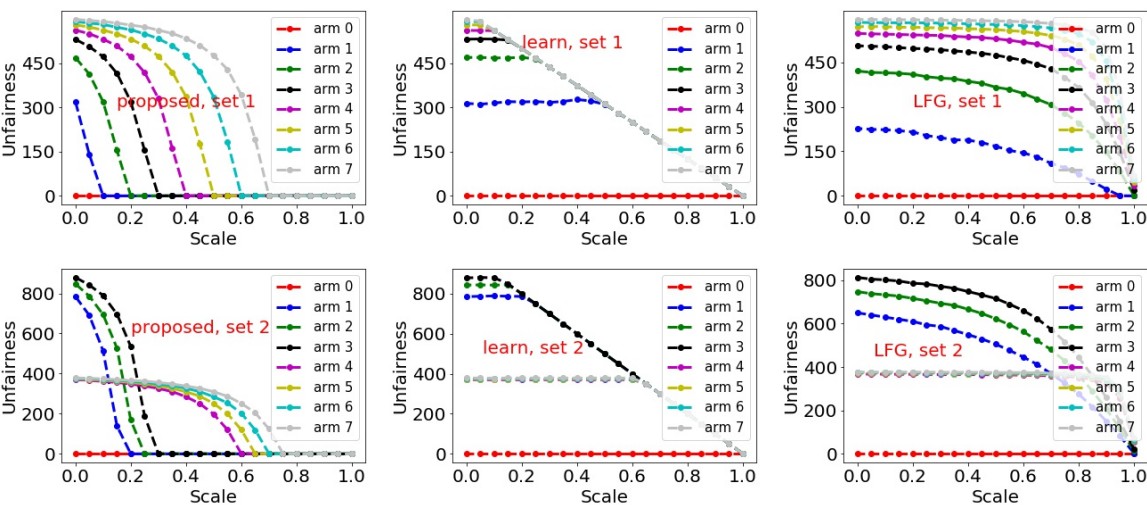

Figure 3: Unfairness path $((\tau_k T - N_k(T))_+, k \in [K])$ for three algorithms under two settings described in Experiment D. (Upper row is for Setting 1 and bottom row is for Setting 2.) For sub-optimal arms, the proposed method can guarantee the fairness with a wider range of tuning parameter. By contrast, Flearn and LFG can break the fairness easily.

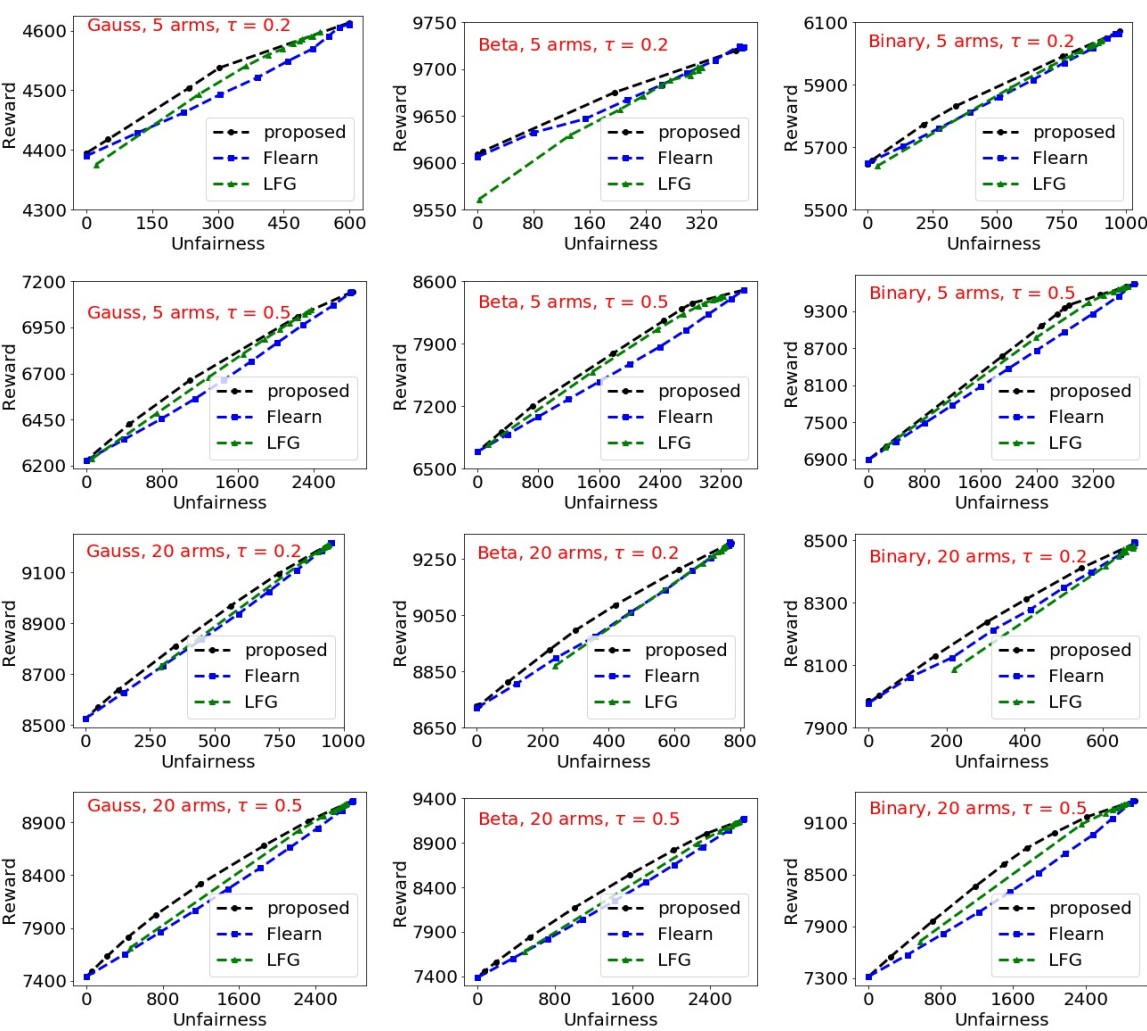

Figure 4: Total regret vs unfairness level for three algorithms under different settings described in Experiment E. (The first row is for 5 arms with required fraction of times $\tau = 0.2$; The second row is for 5 arms with required fraction of times $\tau = 0.5$; The third row is for 20 arms with required fraction of times $\tau = 0.2$; The fourth row is for 20 arms with required fraction of times $\tau = 0.5$. The first column is for Gaussian reward distribution; the second column is for Beta distribution; and the third column is for Bernoulli distribution.) Given the fixed unfairness level, the proposed method can have larger total reward than other two methods consistently over all experimental settings.

**Highlight of the proof**. The technical challenge lies in handling the term $(\tau_k T - N_k(T))_+$. (1) Our main task in proving upper bound is to show that, for any $k \in \mathcal{A}_{cr}$, $(\tau_k T - N_k(T))_+$ is $O(1)$ for in gap-dependent setting and it is $\tilde{O}(T^{2/3})$ for gap-independent setting. Unlike the classical UCB algorithm analysis, we cannot bound $N_k(T)$ separately for each arm $k$. Instead, we need to study the relationship between any pair of critical arms and the relationship between critical arm and non-critical arm. A key step is to find a stopping time $n_1$ such that any arm $k \in \mathcal{A}_{cr}$ satisfies $N_k(n_1) \geq \tau_k n_1$. Therefore, between rounds $n_1$ and $T$, the behavior of $(\tau_k T - N_k(T))_+$ can be well controlled. (2) In proving maximal inequality, we need to order $K$ arms according to the values of $\mu_k + A_k$. Then the bound of $\max_{1 \leq t \leq T}(\tau_k t - N_k(T))_+$ can be obtained by a recursive formula (see equation 30) starting from $k = \bar{k}_1$ to $k = k_K$, where $k_1 := \arg\max\{\mu_k + A_k\}$ and $k_K := \arg\min\{\mu_k + A_k\}$.

## B  PROOF OF GAP-DEPENDENT UPPER BOUNDS

**Proof of Theorem** 1 Its proof is essentially same as the proof of first part in Theorem 2 by treating the non-critical set $\mathcal{A}_{\text{non-cr}}$ empty.

**Proof of Theorem 2** We first prove the first part: $\mathbb{E}[(\tau_k T - N_k(T))_+] = O(1)$ for any $k \in \mathcal{A}_{\text{opt}} \cup \mathcal{A}_{\text{cr}}$.

Suppose at time $n$ that a critical arm $k$ is played less than $\tau_k n$. We can prove that the algorithm pulls critical arm $k'$ at time $n$ such that $N_{k'}(n) \geq 8\log T/c_a^2$ and $N_{k'}(n) > \tau_k n$ with vanishing probability. This is because

$$\mathbb{P}\left(A_k + m_k(n) + \sqrt{\frac{2\log n}{N_k(n)}} \leq m_{k'}(n) + \sqrt{\frac{2\log n}{N_{k'}(n)}}\right)$$

$$\leq \mathbb{P}\left(A_k + \mu_k \leq +\mu_k' + 2\sqrt{\frac{2\log n}{N_{k'}(n)}}\right) + \frac{2}{n^2}$$

$$\leq \mathbb{P}\left(A_k - \Delta_k \leq -\Delta_{k'} + 2\sqrt{\frac{2\log n}{N_{k'}(n)}}\right) + \frac{2}{n^2}$$

$$= 2/n^2. \tag{18}$$

By the same reason, the algorithm pulls non-critical arm $k''$ at time $n$ when $N_{k''}(n) \geq 8\log T/c_a^2$ with vanishing probability.

(Observation 3) In other words, it holds with high probability that once a critical arm $k$ is played with proportion less than required level $\tau_k$'s, it must be pulled in next round when all other arms is played with proportion greater than level $\tau_k$'s and is played more than $8\log T/c_a^2$ times.

(Observation 4) It also holds with high probability that once a non-critical arm is played more than $8\log T/c_a^2$, it can be only played when all critical arms are played with frequency more than the required level $\tau_k$'s.

Moreover, we can show that $N_{k'}(n) \geq 8\log T/c_a^2$ at time $n = c_0 T/2$ for each critical arm $k'$. If not, note that $8\log T/c_a^2 \leq \tau_{k'} c_0 T/4$, then $N_k'(n) < \tau_k' n$ for any $n \in \{\lceil c_0 T/4 \rceil, \ldots, \lfloor c_0 T/2 \rfloor\}$. Hence, for any critical arm $k'$ can be played at most $\max\{\tau_{k'} c_0 T/4, 8\log T/c_a^2\}$ times between rounds $c_0 T/2$ and $c_0 T$; every non-critical arm $k''$ can be played at most $8\log T/c_a^2$ times. Then, we must have

$$c_0 T/2 - c_0 T/4 \leq \sum_k \tau_k c_0 T/4 + \sum_k 8\log T/c_a^2.$$

However, the above inequality fails to hold when $T/\log T \geq 16c_0^2 K/c_a^2$. This leads to the contradiction. Thus, we have $N_{k'}(n) \geq 8\log T/c_a^2$ for any critical arm $k'$ at time $n = c_0 T/2$.

Actually, this further gives us that we must have $N_{k'}(n) \geq \lfloor \tau_{k'} n \rfloor$ for all very critical arms at some time $n \in [c_0 T/2, T]$. To see this, we observe the fact that for any arm $\bar{k}$, it will be played with probability less than $\frac{2}{T^2}$ at time $n$ once $N_{\bar{k}}(n) \geq \max\{\tau_{\bar{k}} n, 8\log T/c_a^2\}$ and one critical arm $k'$ is played less than $\tau_{k'} n$. (In other words, this tells us that once arm $\bar{k}$ has been played $\max\{\tau_{\bar{k}} n, 8\log T/c_a^2\}$ times, then it can only be played at time when all very critical arms $k'$s have been played for $\tau_{k'} n$ times or $\lfloor \tau_{\bar{k}} n \rfloor$ jumps by one with probability greater than $1 - 2/T^2$.)

Let $n_1(\geq c_0 T/2)$ be the first ever time such that $N_{k'}(n_1) \geq \lfloor \tau_{k'} n_1 \rfloor$ for all critical arms $k'$s. By straightforward calculation, it gives that $n_1$ must be bounded by

$$n_1 \leq c_0 T/2 + \sum_{k:\text{non-critical}} 8 \log T/c_a^2 + (\sum_{k:\text{critical}} \tau_k)T$$

with probability greater than $1 - 2K/T$. That is, $n_1$ is well defined between $c_0 T$ and $T$. At time $n_1$, we have all critical arms $k'$ such that $N_{k'}(n_1) \geq \tau_{k'} n_1$.

Moreover, we consider the first time $n_2(> n_1)$ such that every non-critical arm $k''$ has been played for at least $8 \log T/c_a^2$ times when $\Delta_{k''} \leq \sqrt{c_a \frac{\log(c_0 T/2)}{16 \log T}}$ (If $c_a > 4$, it automatically holds for any $\Delta_{k''}$). We claim that $n_2 \leq n_1 + c_0 T/2$. This is because, between rounds $n_1$ and $n_2$, the algorithm will choose non-critical arm $k''$ when $N_{k'}(n) \geq \tau_{k'}(n)$ for all critical arms $k'$s and $N_{k''}(n) \leq \log(c_0 T/2)/2\Delta_{k''}^2$. To see this, we know that

$$\mathbb{P}(m_{k'}(n) + \sqrt{\frac{2 \log n}{N_{k'}(n)}} \geq m_{k''}(n) + \sqrt{\frac{2 \log n}{N_{k''}(n)}})$$

$$\leq \mathbb{P}(\mu_k' + 2\sqrt{\frac{2 \log n}{N_{k'}(n)}} \geq \mu_{k''} + \sqrt{\frac{\log n}{N_{k''}(n)}})$$

$$\leq \mathbb{P}(\mu_k' + 2\sqrt{\frac{2 \log T}{\tau_{k'} c_0 T/2}} \geq \mu_{k''} + \sqrt{\frac{\log(c_0 T/2)}{N_{k''}(n)}})$$

$$\leq 2/c_0 T. \tag{19}$$

That is, index of arm $k''$ is larger than $k'$ with high probability.

In other words, for each round between $n_1$ and $n_2$, each critical arm $k'$ can be only pulled at most $\tau_{k'}(n_2 - n_1)$ before every non-critical arm $k''$ has been played for $\min\{8 \log T/c_a^2, \log(c_0 T/2)/2\Delta_{k''}^2\}$. Additionally, each non-critical arm $k''$ can be only played for at most $8 \log T/(\Delta_k - A_k)^2$ with high-probability. Therefore, it must hold that

$$n_2 - n_1 \leq (\sum_k \tau_k)(n_2 - n_1) + \sum_k 8 \log T/(\Delta_k - A_k)^2.$$

However, the above inequality fails to hold when $n_2 - n_1 \geq c_0 T/2$ under assumption that $\Delta_k - A_k \geq \sqrt{\frac{8K \log T}{c_0^2 T}}$. This validates the claim $n_2 \leq n_1 + c_0 T/2$.

Starting from time $n_2$, by the observations 3 and 4, it can be seen that the maximum values of $(\tau_{k'} n - N_{k'}(n))_+$ for any critical arm $k'$ is always bounded by 1 with probability $1 - 2K/T$ ($n \in [n_2, T]$). This completes the proof of the first part.

For the second part, we need to prove $\mathbb{E}[N_k(T)] \leq \max\{\frac{8 \log T}{\Delta_k^2}, \tau_k T\} + O(1)$ for $k \in \mathcal{A}_{\text{cr}}$.

When $\frac{8 \log T}{\Delta_k^2} > \tau_k T$, we can calculate the probability

$$\mathbb{P}(\text{arm } k \text{ is pulled at round } n + 1 \,|\, N_k(n) \geq \frac{8 \log T}{\Delta_k^2})$$

$$\leq \mathbb{P}(i_k(n+1) \geq i_{k^*}(n+1))$$

$$\leq \mathbb{P}(\hat{m}_k(n+1) + \sqrt{\frac{2 \log(n+1)}{N_k(n)}} \geq \hat{m}_{k^*}(n+1) + \sqrt{\frac{2 \log(n+1)}{N_k(n)}})$$

$$\leq 1/n^2 \leq 1/(8 \log T/\Delta_k)^2 \leq 1/(\tau_k T)^2. \tag{20}$$

When $\frac{8\log T}{\Delta_k^2} \le \tau_k T$, we can similarly calculate the probability

$$
\begin{aligned}
&\mathbb{P}(\text{arm } k \text{ is pulled at round } n+1 \,|\, N_k(n) \ge \tau_k T) \\
\le\quad & \mathbb{P}(i_k(n+1) \ge i_{k^*}(n+1)) \\
\le\quad & \mathbb{P}(\hat{m}_k(n+1) + \sqrt{\frac{2\log(n+1)}{N_k(n)}} \ge \hat{m}_{k^*}(n+1) + \sqrt{\frac{2\log(n+1)}{N_k(n)}}) \\
\le\quad & 1/n^2 \le 1/(\tau_k T)^2.
\end{aligned}
\tag{21}
$$

Hence we can easily obtain that $\mathbb{E}[N_k(T)] \le \max\{\frac{8\log T}{\Delta_k^2}, \tau_k T\} + O(1)$ by union bound.

For the third part that $\mathbb{E}[N_k(T)] \le \min\{\frac{8\log T}{(\Delta_k - A_k)^2}, \tau_k T\} + O(1)$ ($k_j \in \mathcal{A}_{\text{non-cr}}$), it follows from the fact that we can treat $\mu_k + A_k$ as new expected reward for arm $k \in \mathcal{A}_{\text{non-cr}}$. Thus the corresponding sub-optimality gap is $\Delta_k - A_k$. The result follow by using standard technique in the classical UCB algorithm. Hence we omit the details here.

Finally, by combining three parts and straightforward calculation, we obtain the desired gap-dependent upper bounds. This concludes the proof.

## C  PROOF OF GAP-DEPENDENT LOWER BOUNDS

**Proof of Theorem 5.**  We consider the following setting, where arm 1 is the optimal arm with a deterministic reward $\Delta$ and arms $k$, $(k \ge 2)$ are sub-optimal arms with reward zero. Let penalty rate $A_k = A$ for all $k \in [K]$ with $\Delta > A$. Assuming that $\frac{8\log T}{(\Delta - A)^2} \le \tau_k T/2$, we construct a lower bound as follows.

We claim that each arm $k \ge 2$ will be played at least $n_1 := \frac{\log T}{(\Delta - A)^2}$ times. If there exists an arm $k_0$ has not been played for $n_1$ times, we then consider the time index $n_a = T/2 + 1 + (K-2)\frac{8\log T}{(\Delta - A)^2} + n_1$. At this time, we have that arm 1 is the arm with largest index since that for each sub-optimal arm $k \ne k_0$, its index will never exceeds $\Delta$ once it has been played $\frac{8\log T}{(\Delta - A)^2}$ times. According to assumption that arm $k_0$ has been played less than $n_1$ times, thus arm 1 is the arm with largest index at time $n_a$.

However, the index of arm 1 at time $n_a$ is never larger than $\sqrt{\frac{2\log T}{T/2}} + \Delta$. The index of arm $k_0$ at time $n_a$ is always larger than $A + \sqrt{\frac{2\log(T/2)}{n_1}}$. It gives

$$
i_1(n_a) \le \sqrt{\frac{2\log T}{T/2}} + \Delta < A + \sqrt{\frac{2\log(T/2)}{n_1}} \le i_{k_0}(n_a),
\tag{22}
$$

which leads to the contradiction of the mechanism of the proposed algorithm. Hence, we have that each sub-optimal should have been played for at least $\frac{\log T}{(\Delta - A)^2}$ times.

**Proof of Theorem 6.**  We consider the another setting, where where arm 1 is the optimal arm with deterministic reward $\Delta_1 + \Delta_2$, arm $k$'s ($k \in \mathcal{A}_{cr}$) are sub-optimal arms with reward being $\Delta_1$ and arm $k$'s ($k \in \mathcal{A}_{non-cr}$) ar sub-optimal arms with reward being $\Delta_2$. Let penalty rate $A_k = A_2$ for all $k \in \mathcal{A}_{cr}$ with $\Delta_2 < A_2$ and penalty rate $A_k = A_1$ for all $k \in \mathcal{A}_{non-cr}$ with $\Delta_1 > A_1$. Assume that $\sum_{k \in \mathcal{A}_{non-cr}} \frac{8\log T}{(\Delta_1 - A_1)^2} + \sum_{k \in \mathcal{A}_{cr}} \frac{8\log T}{\Delta_2^2} < T/2$ and $\tau_k T \le \frac{\log T}{\Delta_2^2}$ for $k \in \mathcal{A}_{cr}$, we then have the following lower bound.

We claim that for each arm $k \in \mathcal{A}_{cr}$ will be played for at least $n_2 := \frac{\log T}{\Delta_2^2}$ times. If not, there will be at least one arm $k_1 \in \mathcal{A}_{cr}$ has been played for less than $n_2$ times. We consider the time stamp, $n_b = T/2 + 1 + \sum_{k \in \mathcal{A}_{non-cr}} \frac{8\log T}{(\Delta_1 - A_1)^2} + \sum_{k \in \mathcal{A}_{cr}; k \ne k_1} \frac{8\log T}{\Delta_2^2} + n_2$. At this time, we have that arm 1 is the arm with the largest index since that for each arm in $\mathcal{A}_{non-cr}$, its index is always smaller than $\Delta_1 + \Delta_2$ once it has been played for $\frac{8\log T}{(\Delta_1 - A_1)^2}$ times. For each arm $k \in \mathcal{A}_{cr}$ ($k \ne k_1$), its index is also smaller than $\Delta_1 + \Delta_2$ once it has been played for $\frac{8\log T}{\Delta_2^2}$ times. According to assumption that arm $k_1$ has been played less than $n_2$ times, thus arm 1 is the arm with largest index at time $n_b$.

However, on other hand, the index of arm 1 at time $n_b$ is never larger than $\sqrt{\frac{2\log T}{T/2}} + \Delta_1 + \Delta_2$. The index of arm $k_1$ is not smaller than $\Delta_1 + \sqrt{\frac{2\log(T/2)}{n_2}}$. It leads to

$$i_1(n_b) \le \sqrt{\frac{2\log T}{T/2}} + \Delta_1 + \Delta_2 \le \Delta_1 + \sqrt{\frac{2\log(T/2)}{n_2}} \le i_2(n_b),$$

this contradicts with arm 1 is arm with largest index at time $n_b$. Hence, any arm in $\mathcal{A}_{cr}$ should be played at least $\frac{\log T}{\Delta_2^2}$ times.

## D    PROOF OF MAXIMAL INEQUALITY (PROOF OF THEOREM 3)

We can order $K$ arms according to the sums $\mu_k + A_k$'s. Specifically, let the order $k_1, k_2, \ldots, k_K$ be defined by

$$\mu_{k_1} + A_{k_1} > \mu_{k_2} + A_{k_2} > \cdots > \mu_{k_K} + A_{k_K}. \tag{23}$$

For simplicity we assume no ties in equation 23. We also assume that $A_k > \Delta_k$ for all $k \in \mathcal{A}_{\mathrm{opt}} \cup \mathcal{A}_{\mathrm{cr}}$.

We now aim to bound expectations of the $\mathbb{E}\max_{t\in[T]}\big(\tau_k t - N_k(t)\big)_+$ for $k \in \mathcal{A}_{\mathrm{opt}} \cup \mathcal{A}_{\mathrm{cr}}$. We will use the ordering of the arms $k_1, k_2, \ldots, k_K$ defined in equation 23. Take any arbitrary $t \in [T]$ and let $k_j \in \mathcal{A}_{\mathrm{opt}} \cup \mathcal{A}_{\mathrm{cr}}$,

$$m_t^{(j)} = \sup\big\{n = 1, \ldots, t : \tau_{k_j} n \le N_{k_j}(n)\big\}. \tag{24}$$

Suppose for a moment that $m_t^{(j)} < t$. We have

$$\big(\tau_{k_j} t - N_{k_j}(t)\big)_+ \le \tau_{k_j} \tag{25}$$
$$+\tau_{k_j}\#\big\{n = m_t^{(j)} + 1, \ldots, t : \tau_{k_d} n > N_{k_d}(n-1) \text{ for some } d = 1, \ldots, j-1\big\}$$
$$+\tau_{k_j}\#\big\{n = m_t^{(j)} + 1, \ldots, t : \tau_{k_d} n \le N_{k_d}(n-1) \text{ for all } d = 1, \ldots, j-1,$$
$$\text{arm } k_j \text{ not pulled at time } n\big\}$$
$$-(1 - \tau_{k_j})\#\big\{n = m_t^{(j)} + 1, \ldots, t : \tau_{k_d} n \le N_{k_d}(n-1) \text{ for all } d = 1, \ldots, j-1,$$
$$\text{arm } k_j \text{ pulled at time } n\big\}$$
$$=\tau_{k_j} + \tau_{k_j}\#\big\{n = m_t^{(j)} + 1, \ldots, t : \tau_{k_d} n > N_{k_d}(n-1) \text{ for some } d = 1, \ldots, j-1\big\}$$
$$-(1 - \tau_{k_j})\#\big\{n = m_t^{(j)} + 1, \ldots, t : \tau_{k_d} n \le N_{k_d}(n-1) \text{ for all } d = 1, \ldots, j-1\big\}$$
$$+\#\big\{n = m_t^{(j)} + 1, \ldots, t : \tau_{k_d} n \le N_{k_d}(n-1) \text{ for all } d = 1, \ldots, j-1,$$
$$\text{arm } k_j \text{ not pulled at time } n\big\}$$
$$=\tau_{k_j} + \#\big\{n = m_t^{(j)} + 1, \ldots, t : \tau_{k_d} n > N_{k_d}(n-1) \text{ for some } d = 1, \ldots, j-1\big\}$$
$$+\#\big\{n = m_t^{(j)} + 1, \ldots, t : \tau_{k_d} n \le N_{k_d}(n-1) \text{ for all } d = 1, \ldots, j-1,$$
$$\text{arm } k_j \text{ not pulled at time } n\big\}$$
$$-(1 - \tau_{k_j})(t - m_t^{(j)}).$$

The final bound is, clearly, also valid in the case $m_t^{(j)} = t$.

Next,

$$\#\big\{n = m_t^{(j)} + 1, \ldots, t : \tau_{k_d} n > N_{k_d}(n-1) \text{ for some } d = 1, \ldots, j-1\big\} \tag{26}$$
$$=\sum_{d=1}^{j-1}\#\big\{n = m_t^{(j)} + 1, \ldots, t : \tau_{k_d} n > N_{k_d}(n-1), \tau_{k_m} n \le N_{k_m}(n-1), m = 1, \ldots, d-1\big\}.$$

For $d = 1, \ldots, j-1$ denote

$$m_t^{(j,d)} = \sup\big\{n = m_t^{(j)}, \ldots, t : \tau_{k_d} n > N_{k_d}(n-1)\big\}. \tag{27}$$

Suppose, for a moment, that $m_t^{(j,d)} > m_t^{(j)}$. Then

$$
\begin{aligned}
0 < \tau_{k_d} m_t^{(j,d)} - N_{k_d}\big(m_t^{(j,d)} - 1\big) &= \tau_{k_d} m_t^{(j)} - N_{k_d}\big(m_t^{(j)} - 1\big) \\
&\quad + \tau_{k_d}\Big(m_t^{(j,d)} - m_t^{(j)} - \#\big\{n = m_t^{(j)} + 1, \dots, m_t^{(j,d)} : \text{ arm } k_d \text{ pulled}\big\}\Big) \\
&\quad - (1 - \tau_{k_d})\#\big\{n = m_t^{(j)} + 1, \dots, m_t^{(j,d)} : \text{ arm } k_d \text{ pulled}\big\} \\
&= \tau_{k_d} m_t^{(j)} - N_{k_d}\big(m_t^{(j)} - 1\big) + \tau_{k_d}\big(m_t^{(j,d)} - m_t^{(j)}\big) \\
&\quad - \#\big\{n = m_t^{(j)} + 1, \dots, m_t^{(j,d)} : \text{ arm } k_d \text{ pulled}\big\}.
\end{aligned}
$$

We conclude that

$$
\begin{aligned}
&\#\big\{n = m_t^{(j)} + 1, \dots, m_t^{(j,d)} : \text{ arm } k_d \text{ pulled}\big\} \\
&\leq \max_{n=1,\dots,t}\big(\tau_{k_d} n - N_{k_d}(n)\big)_+ + \tau_{k_d}\big(m_t^{(j,d)} - m_t^{(j)}\big).
\end{aligned}
$$

Therefore,

$$
\begin{aligned}
&\#\big\{n = m_t^{(j)} + 1, \dots, t : \tau_{k_d} n > N_{k_d}(n-1),\ \tau_{k_m} n \leq N_{k_m}(n-1),\ m = 1, \dots, d-1\big\} \\
&= \#\big\{n = m_t^{(j)} + 1, \dots, m_t^{(j,d)} : \tau_{k_d} n > N_{k_d}(n-1),\ \tau_{k_m} n \leq N_{k_m}(n-1),\ m = 1, \dots, d-1\big\} \\
&\leq \#\big\{n = m_t^{(j)} + 1, \dots, m_t^{(j,d)} : \text{ arm } k_d \text{ pulled}\big\} \\
&\quad + \#\big\{n = m_t^{(j)} + 1, \dots, t : \tau_{k_d} n > N_{k_d}(n-1),\ \tau_{k_m} n \leq N_{k_m}(n-1),\ m = 1, \dots, d-1, \\
&\hspace{9cm} \text{arm } k_d \text{ not pulled}\big\}
\end{aligned}
$$

$$
\begin{aligned}
&\leq \max_{n=1,\dots,t}\big(\tau_{k_d} n - N_{k_d}(n)\big)_+ + \tau_{k_d}\big(t - m_t^{(j)}\big) \\
&\quad + \#\big\{n = m_t^{(j)} + 1, \dots, t : \tau_{k_d} n > N_{k_d}(n-1),\ \tau_{k_m} n \leq N_{k_m}(n-1),\ m = 1, \dots, d-1, \\
&\hspace{9cm} \text{arm } k_d \text{ not pulled}\big\},
\end{aligned}
$$

and the final bound is clearly valid even if $m_T^{(j,d)} = m_T^{(j)}$. Substituting this bound into equation 26 we obtain

$$
\begin{aligned}
&\#\big\{n = m_t^{(j)} + 1, \dots, t : \tau_{k_d} n > N_{k_d}(n-1) \text{ for some } d = 1, \dots, j-1\big\} \\
&\leq \big(t - m_t^{(j)}\big) \sum_{d=1}^{j-1} \tau_{k_d} + \sum_{d=1}^{j-1} \max_{t'=1,\dots,t}\big(\tau_{k_d} t' - N_{k_d}(t')\big)_+ \\
&\quad + \sum_{d=1}^{j-1} \#\big\{n = m_t^{(j)} + 1, \dots, t : \tau_{k_d} n > N_{k_d}(n-1),\ \tau_{k_m} n \leq N_{k_m}(n-1),\ m = 1, \dots, d-1, \\
&\hspace{9cm} \text{arm } k_d \text{ not pulled}\big\},
\end{aligned}
$$

Substituting this bound into equation 25 gives us

$$
\begin{aligned}
\big(\tau_{k_j} t - N_{k_j}(t)\big)_+ &\leq \tau_{k_j} \tag{28} \\
&\quad + \sum_{d=1}^{j-1} \max_{t'=1,\dots,t}\big(\tau_{k_d} t' - N_{k_d}(t')\big)_+ - \big(t - m_t^{(j)}\big)\left(1 - \sum_{d=1}^{j} \tau_{k_d}\right) \\
&\quad + \sum_{d=1}^{j} \#\big\{n = m_t^{(j)} + 1, \dots, t : \tau_{k_d} n > N_{k_d}(n-1), \\
&\hspace{4cm} \tau_{k_m} n \leq N_{k_m}(n-1),\ m = 1, \dots, d-1,\ \text{arm } k_d \text{ not pulled}\big\} \\
&\leq \tau_{k_j} + \sum_{d=1}^{j-1} \max_{t'=1,\dots,t}\big(\tau_{k_d} t' - N_{k_d}(t')\big)_+ \\
&\quad + \sum_{d=1}^{j} \#\big\{n = 1, \dots, t : \tau_{k_d} n > N_{k_d}(n-1), \\
&\hspace{4cm} \tau_{k_m} n \leq N_{k_m}(n-1),\ m = 1, \dots, d-1,\ \text{arm } k_d \text{ not pulled}\big\}.
\end{aligned}
$$

Taking the maximum over $t$ on both sides of above inequality, we then have

$$\max_{t=1,\ldots,T} \left(\tau_{k_j} t - N_{k_j}(t)\right)_+ \leq \tau_{k_j} + \sum_{d=1}^{j-1} \max_{t=1,\ldots,T} \left(\tau_{k_d} t - N_{k_d}(t)\right)_+ \tag{29}$$

$$+ \sum_{d=1}^{j} \#\big\{n = 1,\ldots,T : \tau_{k_d} n > N_{k_d}(n-1),$$

$$\tau_{k_m} n \leq N_{k_m}(n-1),\ m = 1,\ldots,d-1,\ \text{arm } k_d \text{ not pulled}\big\}.$$

Therefore, we arrive at

$$\mathbb{E}\left(\max_{t=1,\ldots,T} \left(\tau_{k_j} t - N_{k_j}(t)\right)_+\right) \tag{30}$$

$$\leq \tau_{k_j} + \mathbb{E}\left(\sum_{d=1}^{j-1} \max_{t=1,\ldots,T} \left(\tau_{k_d} t - N_{k_d}(t)\right)_+\right)$$

$$+ \sum_{d=1}^{j} \mathbb{E}\left(\sum_{n=1}^{T} \mathbf{1}\big(\tau_{k_d} n > N_{k_d}(n-1),\right.$$

$$\left. \tau_{k_m} n \leq N_{k_m}(n-1),\ m = 1,\ldots,d-1,\ \text{arm } k_d \text{ not pulled}\big)\right).$$

We will prove that for $k_d \in \mathcal{A}_{\text{opt}} \cup \mathcal{A}_{\text{cr}}$

$$\mathbb{E}\left(\sum_{n=1}^{T} \mathbf{1}\big(\tau_{k_d} n > N_{k_d}(n-1),\right. \tag{31}$$

$$\left. \tau_{k_m} n \leq N_{k_m}(n-1),\ m = 1,\ldots,d-1,\ \text{arm } k_d \text{ not pulled}\big)\right)$$

$$\leq b_d \log T + O(1)$$

for $b_d > 0$ that we will compute. It is elementary that $k_j \in \mathcal{A}_{\text{opt}} \cup \mathcal{A}_{\text{cr}}$ implies $k_d \in \mathcal{A}_{\text{opt}} \cup \mathcal{A}_{\text{cr}}$ for $d = 1,\ldots,j-1$. Therefore, it will follow from equation 31, equation 30 and a simple inductive argument that for any $k_j \in \mathcal{A}_{\text{opt}} \cup \mathcal{A}_{\text{cr}}$,

$$\mathbb{E}\left(\max_{t=1,\ldots,T} \left(\tau_{k_j} t - N_{k_j}(t)\right)_+\right) \leq a_j \log T + O(1) \tag{32}$$

with $a_1 = b_1$ and for $j > 1$,

$$a_j = \sum_{d=1}^{j-1} a_d + \sum_{d=1}^{j} b_d,$$

which means that

$$a_j = \sum_{d=1}^{j} (j - d + 1) b_d. \tag{33}$$

We now prove equation 31. We have

$$E_\pi\left(\sum_{n=1}^{T} \mathbf{1}\big(\tau_{k_d} n > N_{k_d}(n-1), \tau_{k_m} n \leq N_{k_m}(n-1),\ m = 1,\ldots,d-1,\ \text{arm } k_d \text{ not pulled}\big)\right)$$

$$= \sum_{m=1}^{d-1} E_\pi\left(\sum_{n=1}^{T} \mathbf{1}\big(\tau_{k_d} n > N_{k_d}(n-1), \tau_{k_m} n \leq N_{k_m}(n-1),\ \text{arm } k_m \text{ is pulled at time } n\big)\right)$$

$$+ \sum_{m=d+1}^{K} E_\pi\left(\sum_{n=1}^{T} \mathbf{1}\big(\tau_{k_d} n > N_{k_d}(n-1),\ \text{arm } k_m \text{ is pulled at time } n\big)\right).$$

Observe that a "no-tie" assumption imposed at the beginning of the section implies that

$$\mu_{k_d} + A_{k_d} > \mu_* \geq \mu_{k_m}.$$

Therefore, we can use once again the usual UCB-type argument to see that for any $m = 1, \ldots, d-1$, for any $B > 0$,

$$E_\pi \left( \sum_{n=1}^{T} \mathbf{1}\left( \tau_{k_d} n > N_{k_d}(n-1), \tau_{k_m} n \leq N_{k_m}(n-1), \text{ arm } k_m \text{ is pulled at time } n \right) \right)$$

$$\leq B \log T + \sum_{n=1}^{T} P_\pi \left( N_{k_m}(n-1) > B \log T, \right.$$

$$\left. \tau_{k_d} n > N_{k_d}(n-1), \tau_{k_m} n \leq N_{k_m}(n-1), \text{ arm } k_m \text{ is pulled at time } n \right)$$

$$\leq B \log T + \sum_{n=1}^{T} P_\pi \left( N_{k_m}(n-1) > B \log T, \right.$$

$$\left. \tau_{k_d} n > N_{k_d}(n-1), \tau_{k_m} n \leq N_{k_m}(n-1), i_{k_m}(n) \geq i_{k_d}(n) \right)$$

$$\leq B \log T + \sum_{n=1}^{T} P_\pi \left( N_{k_m}(n-1) > B \log T, \hat{m}_{k_m}(n-1) + \sqrt{\frac{2 \log n}{N_{k_m}(n-1)}} \right.$$

$$\left. \geq \hat{m}_{k_d}(n-1) + A_{k_d} + \sqrt{\frac{2 \log n}{N_{k_d}(n-1)}} \right).$$

By carefully choosing

$$B = \frac{8}{(\mu_{k_d} + A_{k_d} - \mu_{k_m})^2},$$

we obtain the bound

$$E_\pi \left( \sum_{n=1}^{T} \mathbf{1}\left( \tau_{k_d} n > N_{k_d}(n-1), \tau_{k_m} n \leq N_{k_m}(n-1), \text{ arm } k_m \text{ is pulled at time } n \right) \right) \qquad (34)$$

$$\leq \frac{8}{(\mu_{k_d} + A_{k_d} - \mu_{k_m})^2} \log T + O(1),$$

$m = 1, \ldots, d-1$. The same argument shows that for every $m = d+1, \ldots, K$,

$$E_\pi \left( \sum_{n=1}^{T} \mathbf{1}\left( \tau_{k_d} n > N_{k_d}(n-1), \text{ arm } k_m \text{ is pulled at time } n \right) \right) \qquad (35)$$

$$\leq \frac{8}{(\mu_{k_d} + A_{k_d} - \mu_{k_m} - A_{k_m})^2} \log T + O(1).$$

Now equation 34 and equation 35 imply equation 31 with

$$b_d = \sum_{m=1}^{d-1} \frac{8}{(\mu_{k_d} + A_{k_d} - \mu_{k_m})^2} + \sum_{m=d+1}^{K} \frac{8}{(\mu_{k_d} + A_{k_d} - \mu_{k_m} - A_{k_m})^2}. \qquad (36)$$

Now it follows from equation 36 and equation 33 that for every $j$ such that $k_j \in \mathcal{A}_{\text{opt}} \cup \mathcal{A}_{\text{cr}}$,

$$a_j = 8 \sum_{d=1}^{j} (j - d + 1) \left( \sum_{m=1}^{d-1} \frac{1}{(\mu_{k_d} + A_{k_d} - \mu_{k_m})^2} + \sum_{m=d+1}^{K} \frac{1}{(\mu_{k_d} + A_{k_d} - \mu_{k_m} - A_{k_m})^2} \right).$$

$$(37)$$

We conclude by equation 32 that every $j$ such that $k_j \in \mathcal{A}_{\text{opt}} \cup \mathcal{A}_{\text{cr}}$,

$$E_\pi \left( \tau_{k_j} T - N_{k_j}(T) \right)_+ \leq a_j \log T + O(1), \qquad (38)$$

with $a_j$ given in equation 37.

**Remark.** In the proof, we assume that there is no tie, i.e., $A_{k_{j_1}} + \mu_{k_{j_1}} \neq A_{k_{j_2}} + \mu_{k_{j_2}}$ for any $j_1 \neq j_2 \in [K]$. This assumption is not restrictive since the probability that event "$A_{k_{j_1}} + \mu_{k_{j_1}} \neq A_{k_{j_2}} + \mu_{k_{j_2}}$ for some $j_1 \neq j_2 \in [K]$." is zero when we pick penalty rates $A_k$'s uniformly randomly.

# E    PROOF OF GAP-INDEPENDENT UPPER BOUNDS

**Proof of Lemma 1** We first prove that the algorithm pulls arm $k'$ with $A_{k'} - \Delta_{k'} \leq 2\sqrt{\frac{2\log T}{T^{2/3}}}$ at time $n$ when $N_{k'}(n) \geq T^{2/3}$ and $N_k(n) < \tau_k n$ with vanishing probability. This is because

$$\mathbb{P}(A_k + m_k(n) + \sqrt{\frac{2\log n}{N_k(n)}} \leq A_{k'} + m_{k'}(n) + \sqrt{\frac{2\log n}{N_{k'}(n)}})$$

$$\leq \quad \mathbb{P}(A_k + \mu_k \leq A'_k + \mu'_k + 2\sqrt{\frac{2\log n}{N_{k'}(n)}}) + \frac{2}{n^2}$$

$$\leq \quad \mathbb{P}(A_k - \Delta_k \leq A'_k - \Delta_{k'} + 2\sqrt{\frac{2\log n}{N_{k'}(n)}}) + \frac{2}{n^2}$$

$$= \quad 2/n^2. \tag{39}$$

Next, we say arm $k$ is a very critical arm if arm $k$ satisfies $A_k - \Delta_k \geq 2\sqrt{\frac{2\log T}{T^{2/3}}}$. Otherwise $k$ is a non-very critical arm. In other words, each non-very critical arm can be only played at most $O(T^{2/3})$ times with high probability.

Furthermore, we can show that $N_{k'}(n) \geq T^{2/3}$ at time $n = c_0 T/2$ for each very critical arm $k'$. If not, note that $T^{2/3} \leq \tau_{k'} c_0^2 T/4$, then $N'_k(n) < \tau'_k n$ for any $n \in \{\lceil c_0^2 T/4 \rceil, \ldots, \lfloor c_0 T/2 \rfloor\}$. Hence, for any arm $k''$ can be played at most $\max\{\tau_{k''} c_0 T/2, T^{2/3}\}$ times between rounds $c_0^2 T/4$ and $c_0 T/2$. Then, we must have

$$c_0 T/2 - c_0^2 T/4 \leq \sum_k \tau_k c_0 T/2 + \sum_k T^{2/3}.$$

However, the above inequality fails to hold when $T \geq (4K/c_0^2)^3$. This leads to the contradiction. Thus, we have $N_{k'}(n) \geq T^{2/3}$ for any very critical arm $k'$ at time $n = c_0 T/2$.

This further gives us that we must have $N_{k'}(n) \geq \lfloor \tau_{k'} n \rfloor$ for all very critical arms at some time $n \in [c_0 T, T]$. To prove this, we observe the fact that for any arm $\bar{k}$, it will be played with probability less than $\frac{2}{T^2}$ at time $n$ once $N_{\bar{k}}(n) \geq \max\{\tau_{\bar{k}} n, T^{2/3}\}$ and one critical arm $k'$ is played less than $\tau_{k'} n$. (In other words, this tells us that once arm $\bar{k}$ has been played $\max\{\tau_{\bar{k}} n, T^{2/3}\}$ times, then it can only be played at time when all very critical arms $k'$s have been played for $\tau_{k'} n$ times or $\lfloor \tau_{\bar{k}} n \rfloor$ jumps by one with probability greater than $1 - 2/n^2$.)

Let $n_1 (\geq c_0 T/2)$ be the first ever time such that $N_{k'}(n_1) \geq \lfloor \tau_{k'} n_1 \rfloor$. By straightforward calculation, it gives that $n_1$ must be bounded by

$$n_1 \leq c_0 T/2 + \sum_{k'':\text{non-very critical}} T^{2/3} + (\sum_{k'} \tau_{k'}) T \leq (c_0 + \sum_{k'} \tau_{k'}) T$$

with probability greater than $1 - 2K/T$.

That is, $n_1$ is well defined between $c_0 T/2$ and $T$. At time $n_1$, we have all very critical arms $k'$ such that $N_{k'}(n_1) \geq \tau_{k'} n_1$. Therefore, starting from time $n_1$, the maximum difference between any non-fairness level $(\tau_{k'} n - N_{k'}(n))_+$'s with $k'$ in the set of very-critical arms is always bounded by 1 with probability $1 - 2K/T$ for all $n \in [n_1, T]$.

Lastly, suppose $n_2$ be the last time that arm $k$ is above fairness level. We know at time $n = n_2$, each very critical arm $k'$ is played for at least $\tau_{k'} n_2 - 1$. by previous argument. Then in the remaining $T - n_2$ rounds, we know that each very critical arm is played at most $\tau_{k'} T - \tau_{k'} n_2 + 1$. Then we must have

$$T - n_2 \leq (\sum_{k':\text{very critical}} \tau_{k'})(T - n_2) + K + \sum_{k:\text{non-very critical}} T^{2/3},$$

which implies $T - n_2 \leq (KT^{2/3} + K)/c_0$. This finally implies that $N_k(T) \geq N_k(n_2) \geq \tau_k T - \tau_k(KT^{2/3} + K)/c_0 - 1$ with probability at least $1 - 2K/T$. That is, $\mathbb{E}[(\tau_k T - N_k(T))_+] = \tau_k(KT^{2/3} + K)/c_0 + 1 = O(\tau_k KT^{2/3})$.

We prove the gap-independent upper bound (Theorem 4) by considering the following situations.

**Situation 1.a** For arm $k \in \mathcal{A}_{\text{non-cr}}$ and $\Delta_k \leq 4\sqrt{\frac{\log T}{T}}$, the regret on arm $k$ is upper bounded by

$$(\Delta_k - A_k)(\tau_k T - N_k(T)) \tag{40}$$

if $0 \leq N_k(T) \leq \tau_k T$; or bounded by

$$\Delta_k(N_k(T) - \tau_k T) + (\Delta_k - A_k)\tau_k T \tag{41}$$

if $N_k(T) \geq \tau_k T$.

**Situation 1.b** For arm $k \in \mathcal{A}_{\text{non-cr}}$ and $\Delta_k > 4\sqrt{\frac{\log T}{T}}$,

- if $\Delta_k - A_k > 4\sqrt{\log T / \tau_k T}$ the regret on arm $k$ is upper bounded by

$$(\Delta_k - A_k)\left(\frac{8\log T}{(\Delta_k - A_k)^2} + O(1)\right). \tag{42}$$

- if $\Delta_k - A_k \leq 4\sqrt{\log T / \tau_k T}$ the regret on arm $k$ is upper bounded by

$$(\Delta_k - A_k)\tau_k T + \Delta_k\left[\left(\frac{8\log T}{\Delta_k^2} - \tau_k T\right)_+ + O(1)\right]. \tag{43}$$

In other words, for any arm $k \in \mathcal{A}_{\text{non-cr}}$, its regret is always bounded by

$$4\sqrt{\frac{\log T}{T}}\tau_k T + 4\sqrt{\tau_k T \log T} + 4\sqrt{\frac{\log T}{T}}(N_k(T) - \tau_k T)_+. \tag{44}$$

**Situation 2** We then split set $\mathcal{A}_{\text{opt}} \cup \mathcal{A}_{\text{cr}}$ into two subsets, $\mathcal{A}_{\text{cr, large}}$ and $\mathcal{A}_{\text{cr,small}}$, where

$$\mathcal{A}_{\text{cr, large}} := \{k : A_k - \Delta_k > 4\sqrt{\frac{2\log T}{T^{2/3}}}\}$$

and

$$\mathcal{A}_{\text{cr, small}} := \{k : A_k - \Delta_k \leq 4\sqrt{\frac{2\log T}{T^{2/3}}}\}.$$

For arm $k \in \mathcal{A}_{\text{cr,large}}$, we have $\mathbb{E}[(\tau_k T - N_k(T))_+] = O(\tau_k K T^{2/3})$ by Lemma 1. The regret on arm $k$ is then bounded by

$$\begin{aligned}
&\Delta_k \mathbb{E}[N_k(T) - \tau_k T] + O(A_k \tau_k K T^{2/3}) \\
\leq \quad &\Delta_k \min\{\frac{8\log T}{\Delta_k^2} - \tau_k T, N_k(T) - \tau_k T\} + O(A_k \tau_k K T^{2/3}).
\end{aligned} \tag{45}$$

For arm $k \in \mathcal{A}_{\text{cr,small}}$, the regret on arm $k$ is then bounded by

$$(A_k - \Delta_k)(\tau_k T - N_k(T)) \leq 4\tau_k T^{2/3}(\log T)^{1/2} \tag{46}$$

if $0 \leq N_k(T) \leq \tau_k T$, or

$$\Delta_k \min\{\frac{8\log T}{\Delta_k^2} - \tau_k T + O(1), N_k(T) - \tau_k T\} \tag{47}$$

if $N_k(T) \geq \tau_k T$.

In summary, for any arm $k \in \mathcal{A}_{\text{opt}} \cup \mathcal{A}_{\text{cr}}$,

$$\Delta_k \min\{\frac{8\log T}{\Delta_k^2} - \tau_k T, N_k(T) - \tau_k T\} + O(\max\{A_k \tau_k K T^{2/3}, 4\tau_k T^{2/3}(\log T)^{1/2}\}). \tag{48}$$

Combining above situations, the total regret is upper bounded by

$$\sum_{k\in\mathcal{A}_{\text{non-cr}}} \max\{8\sqrt{\tau_k T\log T} + 4\sqrt{\frac{\log T}{T}}(N_k(T) - \tau_k T)_+\}$$

$$+ \sum_{k\in\mathcal{A}_{\text{opt}}\cup\mathcal{A}_{\text{cr}}} \Delta_k \min\{\frac{8\log T}{\Delta_k^2} - \tau_k T, N_k(T) - \tau_k T\} + O(\max\{A_k\tau_k KT^{2/3}, 4\tau_k T^{2/3}(\log T)^{1/2}\})$$

$$\leq\ 8\sqrt{T\log T}(\sum_{k\in\mathcal{A}_{\text{non-cr}}}\sqrt{\tau_k}) + A_{max}KT^{2/3}(\log T)^{1/2}(\sum_{k\in\mathcal{A}_{\text{cr}}\cup\mathcal{A}_{opt}}\tau_k) + 4\sqrt{\frac{\log T}{T}}\sum_{k\in\mathcal{A}_{\text{non-cr}}}(N_k(T) - \tau_k T)_+$$

$$+ \sum_{k\in\mathcal{A}_{\text{opt}}\cup\mathcal{A}_{\text{cr}}}\sqrt{8(N_k(T) - \tau_k T)_+\log T}$$

$$\leq\ 8\sqrt{T\log T}(\sum_k\sqrt{\tau_k}) + A_{max}KT^{2/3}(\log T)^{1/2} + 4\sqrt{\frac{\log T}{T}}(1 - \tau_{min})T + \sqrt{8\log T}\sqrt{KT(1 - \tau_{min})}$$

$$,$$
$$(49)$$

where 49 uses the fact that $\sum_{k\in\mathcal{A}_{\text{cr}}\cup\mathcal{A}_{opt}}\tau_k \leq \sum_k \tau_k \leq 1$; $\sum_{k\in\mathcal{A}_{\text{non-cr}}}(N_k(T) - \tau_k T)_+ \leq T(1 - \tau_{min})$ and

$$\sum_{k\in\mathcal{A}_{\text{opt}}\cup\mathcal{A}_{\text{cr}}}\sqrt{(N_k(T) - \tau_k T)_+} \leq \sum_k\sqrt{(N_k(T) - \tau_k T)_+} \leq \sqrt{K\sum_k(N_k(T) - \tau_k T)_+} \leq \sqrt{KT(1 - \tau_{min})}$$

by Jenson's inequality.

# F  PROOF OF GAP-INDEPENDENT LOWER BOUNDS

Consider a $K$-arm setting with $\mu_2 = \mu_3 = \ldots = \mu_K = 0$, $\mu_1 = \Delta$ $(0 < \Delta < 1/2)$, $A_1, A_2, \ldots, A_K > 0$, $\Delta < A_k$ for $k \in [K]$, $\tau_1, \tau_2, \ldots, \tau_K \in [0, 1]$.

Since $\sum_{k=2}^{T} N_k(T) \leq T$, then it holds $\mathbb{E}_\pi[N_{k_1}(T)] \leq T/(K - 1)$ with $k_1 = \arg\min_{k>1} \mathbb{E}_\pi[N_k(T)]$ for any policy $\pi$. We then construct another $K$-arm setting with $\mu_{k_1} = 2\Delta$ and all other parameters remain the same.

For policy $\pi$, the regret of the first setting is

$$R_{1,\pi}(T) \geq A\mathbb{E}[(\tau_1 T - N_1(T))_+] + \{\Delta\mathbb{E}[N_{k_1}(T) - \tau_{k_1}T] + A\mathbb{E}(\tau_{k_1}T - N_{k_1}(T))_+\}$$

and the regret of the second setting is

$$R_{2,\pi}(T) \geq A\mathbb{E}[(\tau_{k_1}T - N_{k_1}(T))_+] + \{\Delta\mathbb{E}[N_1(T) - \tau_1 T] + A\mathbb{E}(\tau_1 T - N_1(T))_+\}$$

If $N_1(T) < (1 + \tau_1 - \tau_{k_1})T/2$, then $R_{1,\pi}(T) \geq \Delta\frac{1 - \tau_1 - \tau_{k_1}}{2}T$. While $N_1(T) > (1 + \tau_1 - \tau_{k_1})/2$, then $R_{2,\pi}(T) \geq \Delta\frac{1 - \tau_1 - \tau_{k_1}}{2}T$. In other words, for policy $\pi$,

$$\text{worst regret}\ \geq\ \frac{1}{2}(R_{1,\pi}(T) + R_{2,\pi}(T))$$

$$\geq\ \frac{1}{2}(\Delta T\frac{1 - \tau_1 - \tau_{k_1}}{2}\mathbb{P}(N_1(T) < \frac{1 + \tau_1 - \tau_{k_1}}{2}) + \Delta T\frac{1 - \tau_1 - \tau_{k_1}}{2}\mathbb{P}(N_1(T) \geq \frac{1 + \tau_1 - \tau_{k_1}}{2}))$$

$$\geq\ \frac{(1 - \tau_1 - \tau_{k_1})\Delta T}{8}\exp\{-KL(P_1\|P_2)\}$$
$$(50)$$

$$\geq\ \frac{(1 - \tau_1 - \tau_{k_1})\Delta T}{8}\exp\{-CT\Delta^2/(K - 1)\},$$
$$(51)$$

where $P_1$ and $P_2$ are two probability distributions under two settings associated with policy $\pi$; 50 follows from the Bretagnolle–Huber inequality. Inequality 51 holds since KL-divergence $KL(P_1\|P_2) \leq CT\Delta^2/(K - 1)$ for many probability distributions. (E.g. $C = 1/2$ if the reward of each arm follows Gaussian distribution with variance 1.)

Taking $\Delta = \sqrt{\frac{K-1}{CT}}$, we have

$$
\begin{aligned}
\text{worst regret} \quad &\geq \quad \frac{(1 - \tau_1 - \max_{k \neq 1} \tau_k)\Delta T}{8} \exp\{-CT\Delta^2/(K-1)\} \\
&\geq \quad \frac{(1 - 2\max_k \tau_k)\sqrt{(K-1)T/C}}{8e},
\end{aligned}
\tag{52}
$$

where $e = \exp\{1\}$. This completes the proof of Theorem 7.

