# OpenReview forum: "A new look at fairness in stochastic multi-armed bandit problems"
_ICLR.cc/2022/Conference — ICLR 2022 Submitted_

### Official Review · Reviewer_rWC1 · 2021-11-02

**Correctness:** 4
**Technical Novelty And Significance:** 2
**Empirical Novelty And Significance:** 2
**Recommendation:** 6
**Confidence:** 4

**Main Review:**

Strength of the paper

- Considering fairness in MAB is an important direction, and the current paper studies the variant where the fairness consideration is put into the objective function as a penalty, in contrast with the previous study that treat it as a hard constraint.

- The paper proposes a hard-threshold UCB-based algorithm with regret analysis and empirical good performance.

Weakness of the paper

- The gap-independent regret upper bound of the algorithm has a large gap comparing to the lower bound. It is unclear from the discussion in the paper whether it is due to the analysis or the algorithm. This indicates that the current result of the paper is still transitional.

- The authors do not discuss why they chose this fairness model while there are other fairness model in the literature. There is no application motivation discussed, so it is unclear to the readers what kind of applications are suitable for their model.

- There is no proof outline in the main text for either the upper bounds or the lower bounds, so it is unclear how the analysis deals with the new challenges in the new setting. There is only one short paragraph on "technical challenges" for the upper bound, but it only says the term $(\tau_k T - N_k(T))_{+}$ should be treated carefully, but does not say anything on how to do it. Even when browsing through the supplementary material, there is no outline highlighting the differences and technical novelty of the analysis. One as to read the 10 page proof in detail to perhaps understand the new technical novelty. Without such explanations, the technical novelty if any will not benefit the readers.


Other issues of the paper:

- Page 2, 3rd paragraph, $N_k t$ should be $N_k(t)$

- Page 2, notations paragraph, the slash between $a \le Cb$ and $a \ge b/C$ is confusing since it looks like a division sign. I guess it means the logical AND.

- Page 3, Equation (4). It is better to put a bracket [ ] after the expectation sign $\mathbb{E}$ and the bracket contains the $( )_+$ inside, otherwise, the subscript $+$ seems to be outside the expectation. This applies also to many other cases in the paper.

- Eq.(5) and the explanation afterwards.
I feel that it is more direct to explain that Eq.(4) is the reward achieved by a policy $\pi$, and the regret is comparing this reward with the optimal reward achieved by a prophet policy that knows $\mu_i$'s. Eq.(5) gives me the impression that it is comparing against always playing the best arm, which is no longer the best policy under the penalty term, and later I realized that it is just a different way to derive the regret I described above.

- Eq.(6) and the equation above.
Why are there still the expectation sign in the equations? There is no longer randomness in the formula I think.

- Page 5, 2nd paragraph, "By choosing the penalty rate properly"
This expression gives me the impression that we could choose the penalty rate, but it should be given as problem input, not chosen, right?

This is related to the condition $A_k - \Delta_k \ge c_a$ in Theorem 1. I would like to see more discussions on whether this assumption is reasonable, and if there is any thing we can do if the assumption does not hold.

- Page 5, Theorem 2
In the line before the third equation in the theorem, notation $k_j$ seems to be wrong. I guess it should be $k$.

Also, there is a redundant "the" in the second line after the theorem.

- Page 8, Experiment A, Case 3. $A$ should be $A_k$?

**Summary Of The Paper:**

The paper studies one variant of fair multi-armed bandit (MAB) problem: For each arm $k$ played, there is a known fairness parameter $\tau_k$, which gives the fraction of times that this arm should be played, and if the total time $k$ is played by time horizon $T$ is less than $\tau_k T$, a penalty $A_k$ is added to the regret term for each insufficient play. The authors propose a hard-threshold UCB-based algorithm, provide its gap-dependent and gap-independent regret bounds, and show that the gap-dependent bound almost matches the lower bound, while the gap-independent bound has a significant difference between the lower bound --- the lower bound is $O(T^{1/2})$ while the current upper bound is $O(T^{2/3})$.


**Summary Of The Review:**

The paper has positive contribution to the community on studying the penalty-based fairness condition in MAB, but due to the weakness pointed out, its contribution is limited and is not easy to understand. Therefore, I think it is not a case of clear acceptance, so I give the marginal accept recommendation.

---

> ### Author Response · Authors · 2021-11-15
> **Response to Reviewer rWC1**
>
> We really appreciate your recognition of our contribution and thank you for your valuable comments.
>
> -- On gap-independent bound:
> The tricky part is dealing with $(\tau_k T - N_k(T))_{+}$. This quantity not only depends on the arm $k$, but also depends on arms other than the $k$-th arm. The classical UCB algorithm analysis only bounds $N_k(T)$ separately for each arm $k$ and therefore cannot be directly applied to our setting.
> Our current analysis can only show $O(T^{2/3})$ upper bound. We believe it is not due to the algorithm, but need to seek a new proof technique in the future work.
>
>
> -- Motivation and applications of model:
> (i) The main difference between other fairness method is that we want to better balance between the total reward and fairness.
> Given the results in Experiment E, we can find that our proposed method can achieve the highest reward under the same unfairness level.
> (ii) For the application, we can consider the following situation.
> A company want to invest on products. Ideally, they want to engage with the market as much as possible to increase its impact on consumers. That is, they want to invest on each potential product with certain market share. However, a very practical strategy is that, a company cannot invest every one of product and it is better to invest on a sub-group of products with higher profits. In such situation, this can be well adapted to our penalized MAB framework. The critical arms correspond to the products with higher profits.
> (iii) Also see our response to the first reviewer.
>
>
> -- No proof outline: we are really sorry for not giving the proof outline.
> We re-emphasize here that the technical challenge is that
> the arm ($:= \arg\max_k \mu_k + A_k \mathbf 1\{N_k < \tau_k n\}$) becomes the ``optimal arm" instead of arm $:= \arg\max_k \mu_k$.
> In other words, the optimal depends on the value of $N_k$ and round $n$. So we need to compare all pairs of $K$ arms rather than to compare between the $K-1$ sub-optimal arms and the optimal arm $\arg\max_k \mu_k$.
> The key of proof is to show the time $n_1 := \min \{n \in [T] | N_k(n) \geq \lfloor \tau_k n \rfloor, \forall k \in \mathcal A_{cr} \}$ exists with high probability. Then for round $n$ between $n_1$ and $T$, the behavior of $(N_k(n) - \tau_k n)_{+}$ can be well controlled.
>
>
> -- Other issues:
>
> Conditions in Theorem 1: (i) We can improve the condition $A_k - \Delta_k \geq c_a$ to $A_k - \Delta_k \geq \sqrt{\frac{\log T}{c_b T}}$ ($c_b$ is an additional constant)
> to ensure the fairness guarantee.
> (ii) The situation that $A_k - \Delta_k$ is between $0$ and $\sqrt{\frac{\log T}{c_b T}}$ is a statistical-indistinguishable regime. That is, it is hard to detect whether an arm is critical or non-critical over $T$ rounds of time.
> (iii) Moreover, we do not expect the asymptotic fairness guarantee for sub-optimal arms with $A_k - \Delta_k < 0$.
>
> Lastly, thank you for pointing out those typos and we will correct them.
> Hope our response can eliminate your concerns.

---

### Official Review · Reviewer_rUcE · 2021-11-02

**Correctness:** 3
**Technical Novelty And Significance:** 3
**Empirical Novelty And Significance:** 2
**Recommendation:** 5
**Confidence:** 4

**Main Review:**

- Strengths:
  -  The idea of the algorithm is quite intuitive, however seems novel (not been proposed and analyzed in the literature).
  -  The analysis is thorough and rigorous.

- Weaknesses:
  - It is crucial to select a penalty $A_i$ larger than the suboptimality gap of arm $i$ ; otherwise, this arm will only get pulled for a number of epochs logarithmic  to the horizon $T$, therefore totally failing the fairness constraint.
  - Comparison with the state of the art is rather weak. First, comparing them with respect to the penalized regret is unfair to the baselines, as they are not optimized against it and the penalties here serve more as a way to ensure the fairness constraints than an end to themselves; A more illustrative comparison would be to compare the "unfairness" and "normal" regret (without the penalties) of each algorithm. Second, when comparing for different values of the scale parameter, $\eta$, the threshold and weight in FLearn and LFG, respectively, both scale with the horizon $T$; this means that, in values of $\eta$ not close to $1$, the parameters of these algorithms are too large. Experiments with a more reasonable range for the parameters of the state of the art need to be presented.

- Additional Comments:
   - It would be nice to have a comparison on the theoretical results on regret (without the penalty) and unfairness of the proposed algorithm and the two baselines (e.g. the scalings with $T$ in some table).
   - It would also be nice to have more details in the figure captions on what the figure is showing - for instance, what is the setting for each row of plots.

**Summary Of The Paper:**

This paper examines the problem of reward maximization in a Multi-Armed Bandit, with the additional fairness constraint that each arm should be pulled for at least a fraction of time. The authors propose and analyze an algorithm where the UCB index of an arm $i$   is increased by a specified amount (which is a parameter for the algorithm) $A_i$ for the time instances where the arm is lagging behind the fairness constraint. Results about the asymptotic satisfaction (or not) of the fairness constraints depending on the relationship between $A_i$ and the suboptimality gap of each arm are given, as well as results about the gap-dependent and gap-independent regret of the studied algorithm under a modified reward with the threshold (that is, a cost $A_i$ is given to all arms $i$ that do not satisfy the fairness constraints at time $t$). This algorithm is also compared with two recent state of the art algorithms for this problem in synthetic examples and illustrates superior performance.

**Summary Of The Review:**

A solid analysis of a novel and intuitive algorithm for the MAB problem under a type of fairness constraints, however the comparisons with related work are rather weak.

---

> ### Author Response · Authors · 2021-11-15
> **Response to Reviewer rUcE**
>
> We thank the reviewer for your important comments.
>
> -- On penalty: In many applications, the reward are bounded (e.g. 0/1 loss). We can just take penalty rate $A_k > 1$ for guaranteeing the fairness.
> More importantly, we are not only interested in guaranteeing the fairness but also focus on balancing between fairness and reward.
> Having penalty, it allows us easily to make trade-off between them.
>
>
> -- Comparison with other works:
> First question: We indeed compare the "unfairness" and "normal" regret (we used expected reward which is equivalent to the normal regret) of each algorithm in Experiment E as reviewer suggested.
> Such plot can be found in the appendix A. It tells us that the proposed algorithm achieves the highest "normal" reward (not the penalized reward) given the same level of "unfairness"
> under different parameter settings with different reward distributions.
>
>
> Second question: The reasons why we choose the current range of scale parameter are given as follows.
>
> (i) In Experiment D, we aim to show the "unfairness" path of each arm under three different methods. To do this, we need to chose eta equally spaced between $[0,1]$ to illustrate what may happen when $\eta$ deviates far away from $1$.
> Based on Figure 3, we can see the intrinsic differences between our method and the existing methods. Our methods shares the similarity with LASSO method and can make arms with small sub-optimality gap meet
> fairness requirement more easily, while other methods cannot do this.
>
> (ii) In Experiment E, as you mentioned, we have already chosen a lot of $\eta$ values close to 1 for LFG and Flearn methods (e.g. $\eta = 1-0.001, 1 - 0.01, 1 - 0.1, ...$), to produce the curve "Reward" vs "Unfairness level" in Figure 4.
>
> Minor points:
> We will add some discussions on theoretical results of two baselines.
> We will make the figure caption more informative.
> Thank you for these suggestion.
> Hope we have addressed your questions properly.

---

> ### Author Response · Authors · 2021-11-30
> **Response to Reviewer rUcE**
>
> Any further comments or questions are welcomed! Thank you for your time!

---

### Official Review · Reviewer_NynE · 2021-11-03

**Correctness:** 3
**Technical Novelty And Significance:** 1
**Empirical Novelty And Significance:** 2
**Recommendation:** 5
**Confidence:** 4

**Main Review:**

Strength:
- The authors propose a novel algorithm for the penalized variant of the fair MAB problem with the tight analyses of the gap-dependent and -independent regrets.

Weakness:
- The definition of the fairness constraint is not reasonable.
- The paper lacks the rigorous guarantee of fairness.


I recommend the rejection of this paper due to the lack of a reasonable fairness guarantee.

First of all, the main issue of this paper is the problem formulation in Eq. 2, particularly its constraints. The constraints require the algorithm to pull each arm at least the specified times ONLY AT THE END OF ROUNDS. Indeed, we can easily come up with a trivial solution to this problem:
1. Pull each arm $\lceil\tau_kT\rceil$ times.
2. Conduct the optimal algorithm for the unconstrained MAP problem.

Unlike Eq. 2, the existing work, Patil et al. 2020, employs the constraint of anytime guarantee of fairness.  That is, the algorithm needs to pull each arm at least the specified fraction of times FOR ANY ROUND.

The MAB with the penalized reward in Eq. 3 might be a proxy for solving the MAB with the anytime guarantee of fairness. That is, the proposed algorithm has a chance to guarantee the anytime guarantee of fairness. However, it is necessary to analyze and demonstrate the rigorous guarantee of fairness.

The claims regarding the lower bound are somewhat incorrect. In the introduction, the authors mention the optimality of the obtained regret. However, what the authors analyzed is the lower bounds on the regrets of the present algorithm and is not the lower bounds on the regrets of any algorithm. Hence, while the lower bounds indicate the tightness of the bounds, they do not reveal the optimality of the bounds.

**Summary Of The Paper:**

The authors deal with the stochastic multi-armed bandit problem with constraints in the least number of arms pulls for fairness. They convert such a problem into the unconstrained but penalized version. Then, they propose the hard-threshold UCB algorithm with the analyses of its gap-dependent and -independent regrets. The lower bounds on these regrets of the present algorithm.

**Summary Of The Review:**

I recommend the rejection of this paper due to the lack of a reasonable fairness guarantee.  See the detailed comment.

---

> ### Author Response · Authors · 2021-11-15
> **Response to Reviewer NynE**
>
> Thank you for your comments. Given below are our detailed responses.
>
> -- The definition of the fairness constraint is not reasonable:
> Thanks for raising this concern.
> From our point of view, our formulation is reasonable.
> (i) What we are trying to optimize is equation (4), the penalized regret, not equation (2).
> So our proposed algorithm is designed for better balance between the fairness and reward.
> The trivial solution mentioned by the reviewer is not optimal in our proposed framework.
> (ii) We admit that Patil et.al. (2020) propose a ANY-ROUND fair MAB method. However, they also consider the regret at the end time $T$, i.e.,
> $R(T) = \sum_{k} \Delta_k \cdot (\mathbb E[N_k(T)] - \max(0, \lfloor \tau_k T\rfloor - \alpha))$.
> In other words, their objective function also only evaluates the performance at $T$. So we think our penalization framework should be OK as well.
> Moreover, their regret is not well-defined. It may happen that the regret could be negative for some policy $\pi$, while our penalized regret applies to any policy $\pi$.
> (iii) In quite many applications, people do not care about ANY-ROUND fairness but care about fairness over a certain period of time.
> For example, a company only need maintaining a good market share at the end of a month (quarter) instead of maintaining the market share at every minute.
>
>
>
> -- The paper lacks the rigorous guarantee of fairness:
> We think the current paper contains enough results for the guarantee of fairness: (i) we have the results of asymptotic fairness at round $T$ (Theorem 1); (ii) we have the results of maximum inequality of unfairness over all $t$ (Theorem 3).
> It says that there is at most of $O(\log T)$ unfairness for any critical arms.
> We do not have ANY-ROUND fairness guarantee. But it is nearly any-round fairness guarantee if we ignore the $\log T$ term.
>
>
> -- Claims on the lower bounds:
> First, we feel sorry to make you confused about our lower bound results. But from our perspective, we do not make incorrect claims on that.
> For example, in abstract, we wrote "Lower bounds are also given to illustrate the tightness of our theoretical analysis".
> This exactly means that our upper bound and lower bound match each other. Our analysis is tight.
> In the introduction, we wrote "The algorithm is also shown to obtain the nearly optimal $O(\log T)$ regret when the sub-optimality
> gap is assumed to be fixed."
> This means that the order $\log T$ is typically the best order we can achieve in the UCB-type algorithm.
> On page 5, we also pointed out that when $A_k \equiv 0$, the bound matches the know result in Auer et.al. 2002.
> That is why we say our result is nearly optimal.
>
> Additionally, "lower bounds on the regrets of the specific algorithm"-type result is also used in the literature (e.g. Lee et.al. 2020, optimal algorithms for stochastic multi-armed bandit with heavy tailed rewards).
> We agree to make the claim clearer in the final version.
> We hope we have addressed all of your concerns.

---

### Official Review · Reviewer_PJx4 · 2021-11-04

**Correctness:** 4
**Technical Novelty And Significance:** 3
**Empirical Novelty And Significance:** 3
**Recommendation:** 5
**Confidence:** 4

**Main Review:**

Strengths:
1. The authors propose a new framework with penalty terms to model fairness constraints, which is different from previous fairness MAB formulations.
2. The proposed penalized regret (Eq.(10)) helps justify the tradeoff between reward and fairness.
3. Both upper and lower regret bounds of the hard-threshold UCB algorithm are provided.
4. Experimental results are consistent with the theoretical findings and show the superiority of the proposed algorithm over other methods.

Weaknesses:
1. Motivation. Although the proposed penalization framework makes sense, it is unclear why we need this new framework. Is it just another way to model the fairness MAB? Does it have any advantages over other formulations? Some motivating examples/applications might be helpful.
2. The idea behind the hard-threshold UCB algorithm seems to be very straightforward: just adding penalty terms to the original UCB values to favor arms that have not satisfied the fairness requirements.
3. For the gap-dependent upper bound in Theorem 2, is it possible to remove the assumption that $\Delta_k \ge 4\sqrt{\frac{\log T}{\tau_k T}}$? Otherwise, I think the regret upper bound without this assumption will depend on $4\sqrt{\frac{\log T}{\tau_k T}}$, then it becomes unclear how $\tau_k$ would affect the regret (i.e., smaller $\tau_k$ might incur larger regret).
4. There is a gap between the upper bound $O(T^{2/3})$ in Theorem 4 and the lower bound $O(\sqrt{T})$ in Theorem 7. Though the authors said it is an open question to improve the gap-independent to be $O(\sqrt{T})$, IMHO, the $O(T^{2/3})$ upper bound is typical for ETC (Explore-Then-Commit) algorithms but not UCB-type algorithms. A natural question that arises: is it possible to modify the ETC algorithm to achieve the same regret bound as the hard-threshold UCB algorithm?
5. The experiments are only running on small synthetic data. It is better to show some experiments with real-world data.


**Summary Of The Paper:**

The authors propose a new penalization framework for the stochastic multi-armed bandit problem with fairness constraints. They formalize the penalized regret to evaluate the performance of bandit algorithms under this setting. They propose a hard-threshold UCB algorithm and provide its gap-dependent/independent regret bounds. Experiments on synthetic data show its superiority over other existing methods.

**Summary Of The Review:**

This paper introduces a new penalization framework for the fairness MAB problem. The authors propose a UCB-type algorithm with theoretical guarantees. From the technical perspective, my main concerns are: 1) relatively simple algorithm design; 2) some issues of the regret analysis. Besides, the motivation for introducing such a new framework and its potential real-world applications are unclear in the current version.

---

> ### Author Response · Authors · 2021-11-15
> **Response to Reviewer PJx4**
>
> Thank you for your comments. Given below are our point-to-point response.
>
> -- About motivation:
> The reasons why we consider the penalization framework are given as follows.
> (i) First, the regret used in Patil et.al. (2020) is not well-defined for all policies.
> (ii) We want to have a better balance between total reward and fairness.
> Based on experiment E, our results are positive and achieve this goal.
> (iii) Our approach to fairness is "soft" in the sense that the output of our algorithm does not guarantee that every arm is pulled the nominally minimal fraction of times. However, we allow the user to specify how "hard" or how "soft" the fairness requirement on each arm is. Selecting a very large penalty $A_k$ for arm k would asymptotically guarantee that the arm is pulled the nominally minimal fraction of times. The advantage of our approach is that it allows the user to distinguish, if desired, between arms for which is more important to sample an arm with the required frequency and those arms for which it is less important to do so.
> (iv) Our new framework can be used in many applications. For example, in finance, a company not only want to maximize their profits but also to have a healthy market share.
> But a realistic question is that the company cannot invest on every products. In other words, they need to determine a subgroup. Our penalized method can find a set of critical arms to help the balance between profits and market share.
>
> -- Idea is simple: From our perspective, a good algorithm should be simple but effective. The proposed hard-threshold UCB algorithm really works well in both theoretical and empirical sense.
> In Section 5 (the comment section), we have pointed out that the hard-threshold term cannot be replaced arbitrarily (i.e. soft-threshold term). Hence we think the proposed method has a decent contribution to the machine learning society, though it is simple.
>
> -- Remove assumption $\Delta_k \geq 4 \sqrt{\frac{\log T}{\tau_k T}}$:
> Thank you for raising this technical question. Yes, we can relax this condition.
> When $\tau_k$ becomes smaller, we can obtain the following bound.
> It holds that
> \begin{eqnarray}
> \mathbb E[N_k(T)] \leq \max\{\min \{\frac{8 \log T}{(\Delta_k - A_k)^2}, \tau_k T\}, \frac{8 \log T}{\Delta_k^2} \} + O(1)
> \end{eqnarray}
> for any $k \in \mathcal A_{non-cr}$. When $\tau_k \rightarrow 0$, the additional term $\frac{8 \log T}{\Delta_k^2}$ is from the classical analysis of UCB algorithm.
> In current version, we do not have this result because we think this is a fairly standard by-product.
> Of course, we will add this in the paper if it could be accepted.
>
>
> -- About ETC algorithm: Thank you for raising this new perspective.
> We believe it is possible to modify Explore-Then-Commit algorithm into our framework. Worse case regret $O(T^{2/3})$ may be possible.
> However, the proposed method has different properties from ETC-type method.
> The proposed one can automatically adjust the unfairness level for distinct arms and can have better trade-off between reward and unfairness.
> We will add this discussion of ETC method into the final version.
>
>
> -- About experiments:
> Since our paper mainly focuses on the new theory and methodology, the current experimental result is already much more comprehensive and informative than that in Patil et al. 2020 (F-learn method).
> We agree to add more results on larger synthetic datasets in the final version.
>
> Hope we have addressed all of your concerns.

---

> > ### Comment · Reviewer_PJx4 · 2021-11-30
> > **Thanks for the response**
> >
> > I appreciate the discussion of motivation. However, I still have concerns about the relatively loose regret upper bound (T^{2/3}) of the UCB-type algorithm, and, as pointed out by another reviewer, the lower bounds specific to the present algorithm. Thus, I would like to maintain my score.

---

> > > ### Author Response · Authors · 2021-11-30
> > > **Response to PJx4 after initial response**
> > >
> > > Dear Reviewer,
> > >
> > > Thank you for your response to our rebuttal and we are happy to see that there is no other doubt other than the model-independent upper bound, as raised in the original review.
> > >
> > > On the other hand, we can see that you have, after reading other reviews, additional comments regarding our bound of UCB-type algorithm.  As we commented in the context and in the response to other reviewers, the contributions of our paper  probably should not be considered  marginal, because
> > >
> > > (1) The penalization framework is fully new which did not appear in any other literature.
> > >
> > > (2) Though algorithm is simple (which we hope is also a merit), the empirical result is much better than baselines.
> > >
> > > (3) The regret analysis is complete. Especially, the analysis of model-dependent bound is tight. The analysis of model-independent upper bound is also non-trivial and more complicated than classical UCB algorithm.
> > >
> > > (4) The current work opens a room for future research on this track.
> > >
> > > Lastly, we thank the reviewer again and appreciate your time!

---

### Decision · Program_Chairs · 2022-01-20

**Decision:**

Reject

**Comment:**

The paper provides an algorithm for the stochastic multi armed bandit (MAB) problem in the regime with fairness constraints. It continues a line of work that in high level define fairness as a requirement to ensure a minimum amount of exploration for every arm.
The main concern I found in the reviews regards the definition of fairness in this paper. Although it follows the same high level narrative of previous works its exact definition and difference from previous papers is not convincingly motivated, and seems to be tailored to the proposed algorithm rather to a real world fairness constraint. This issue could have been mitigated by a novel or generalizable technique, or insightful experiments, but this does not seem to be the case given the reviewers comment about the limited novelty and basic experiments.